# Magnetically assisted drop-on-demand 3D printing of microstructured multimaterial composites

Wing Chung Liu [1], Vanessa Hui Yin Chou[1], Rohit Pratyush Behera [1] & Hortense Le Ferrand [1,2] ✉

Microstructured composites with hierarchically arranged fillers fabricated by three-dimensional (3D) printing show enhanced properties along the fillers' alignment direction. However, it is still challenging to achieve good control of the filler arrangement and high filler concentration simultaneously, which limits the printed material's properties. In this study, we develop a magnetically assisted drop-on-demand 3D printing technique (MDOD) to print aligned microplatelet reinforced composites. By performing drop-on-demand printing using aqueous slurry inks while applying an external magnetic field, MDOD can print composites with microplatelet fillers aligned at set angles with high filler concentrations up to 50 vol%. Moreover, MDOD allows multimaterial printing with voxelated control. We showcase the capabilities of MDOD by printing multimaterial piezoresistive sensors with tunable performances based on the local microstructure and composition. MDOD thus creates a large design space to enhance the mechanical and functional properties of 3D printed electronic or sensing devices using a wide range of materials.

Three-dimensional (3D) printing is a manufacturing technology that generates freeform 3D structures using layer-by-layer deposition. Traditionally, 3D printing is used for small batch prototyping with limited material compatibilities. However, recent advancements have allowed more classes of materials to be 3D printed for use in multidisciplinary fields such as aerospace, robotics, biomedical and electronic applications[1–4]. More recently, 3D printing has been extended to fabricate microstructured composites consisting of orderly arranged 1D fibrous or 2D plate-shaped anisotropic reinforcement fillers. Microstructured composites are interesting as they give rise to superior properties. For example, bioinspired Bouligand and nacre-like structures are formed by stacked layers of aligned stiff fibres and platelets respectively, within a soft matrix. These structures have high stiffness owing to the high filler content, while their layered architecture toughens the structure[5]. Aside from mechanical properties, microstructured composites containing functional fillers such as aligned graphene or hexagonal boron nitride (hBN) microplatelets also

display enhanced thermal and electrical properties[6,7]. Although exciting progress has been made in the field of 3D printing, it is still challenging to fabricate these microstructured composites due to the complex anisotropic and multimaterial layered structure required.

One strategy to print microstructured composites is to use multimaterial methods such as polyjet printing or fused deposition modelling to print combinations of soft and stiff materials[8–10]. However, such techniques are usually limited to printing polymers with varied mechanical properties and cannot fully realise the same properties as composites containing actual stiff fillers. Current methods that utilise inks containing stiff fillers generally make use of shear-induced or field-assisted alignment of the fillers[11,12]. 3D printing of microstructured composites reinforced by 1D fillers is easier to achieve as the fillers only need to be aligned along one direction. Conversely, the fabrication of microstructured 2D platelet-based composites is more demanding as the fillers have an additional axis to align. Despite so, the 3D printing of platelet-based composites has been accomplished in various research

[1]School of Mechanical and Aerospace Engineering, Nanyang Technological University, Singapore 639798, Singapore. [2]School of Materials Science and Engineering, Nanyang Technological University, Singapore 639798, Singapore. ✉e-mail: hortense@ntu.edu.sg

groups. Yang et al. 3D printed nacre-inspired graphene nanoplatelet structures within a photocurable resin using an electric field-assisted stereolithography (SLA) method[13]. Magnetic fields have also been used in similar ways to biaxially align alumina microplatelets in 3D printed photocurable polymers[14,15]. While these methods are effective in printing microstructured materials, the use of photocurable polymer inks limits the filler loadings as high filler loadings lead to high ink viscosities which impede the rotation of the microplatelets during alignment. As a result, the final composites often have low solid loadings of <15 vol% which limits their properties. To fabricate composites with higher solid loadings, Feilden et al. used direct ink writing (DIW) to print ceramic composites using an alumina-based hydrogel ink. The shear forces developed during printing aligned the microplatelets along the circular circumference of the dispensing nozzle[16]. Although samples with high solid contents of ~50 vol% were printed, the filler alignment cannot be freely controlled. Hence, a printing technique that allows the control of filler orientation while maintaining high solid loadings in the printed structures would be highly desirable.

To achieve this goal, the use of solvent-based slurry ink is a promising option. These types of inks generally have lower viscosities than resin-based inks, which allow the fillers to be easily aligned. At the same time, densification occurs as the solvent evaporates, leading to structures with high solid contents[17]. This strategy has been previously used in magnetically assisted slip casting to fabricate high solid content microstructured composites with controllable microplatelet alignment[18]. Using similar aqueous slurries in a drop-on-demand 3D printing technique augmented with magnetic fields for alignment, additional voxelated control in the printed structures could be achieved.

In this work, we apply this principle and develop a 3D printing technique, which we call magnetically assisted drop-on-demand printing (MDOD). MDOD can print microplatelet-based microstructured composites with high filler loadings up to ~50 vol%, and with locally varying filler orientation and composition. To achieve this, aqueous inks with magnetically responsive anisotropic microplatelets

are deposited in droplets onto a substrate while a magnetic field is applied to induce filler alignment. We also show that MDOD can be easily applied to different types of microplatelets, leading to the possibility of multimaterial printing. The material and filler alignment in each individual droplet can be tuned to give voxelated control of the overall printed structure. To demonstrate the advantage of MDOD, multifunctional devices such as piezoresistive sensors are fabricated and their performances and limitations are assessed. By controlling their local microstructure and composition, the mechanical and functional properties are enhanced. The ability to vary the material composition and microstructure creates a large design space to tune the device performance based on the needs of the target applications. The work presented here can be leveraged to fabricate novel microstructured materials and also offer an alternative approach to improve the performances of 3D printed functional devices.

## Results

### Printing strategy

To achieve voxelated and microstructured microplatelet composites with high solid content, MDOD combines magnetic alignment with drop-on-demand printing using slurry inks. These inks consist of magnetically responsive microplatelets dispersed in a solvent with a polymeric binder. The microplatelets are rendered magnetically responsive by functionalisation with superparamagnetic nanoparticles (SPIONS)[19]. During the printing process, a droplet-on-demand printer deposits ink droplets with microplatelet concentrations $\phi_i$ onto a substrate (Fig. 1). Meanwhile, a magnetic field of strength $B$ and rotating at a frequency $f$ above a critical frequency is applied to induce the biaxial alignment of the microplatelets in the plane of the magnetic field rotation. The orientation of the microplatelets, defined by the angle $\theta$ with respect to the plane of the substrate, can be tuned as desired. Sedimentation then occurs which densifies the platelet concentration from $\phi_i$ to a final value $\phi_f$. While sedimentation is typically undesirable for printing inks, sedimentation in MDOD helps to further increase the microplatelets concentration to achieve high platelet

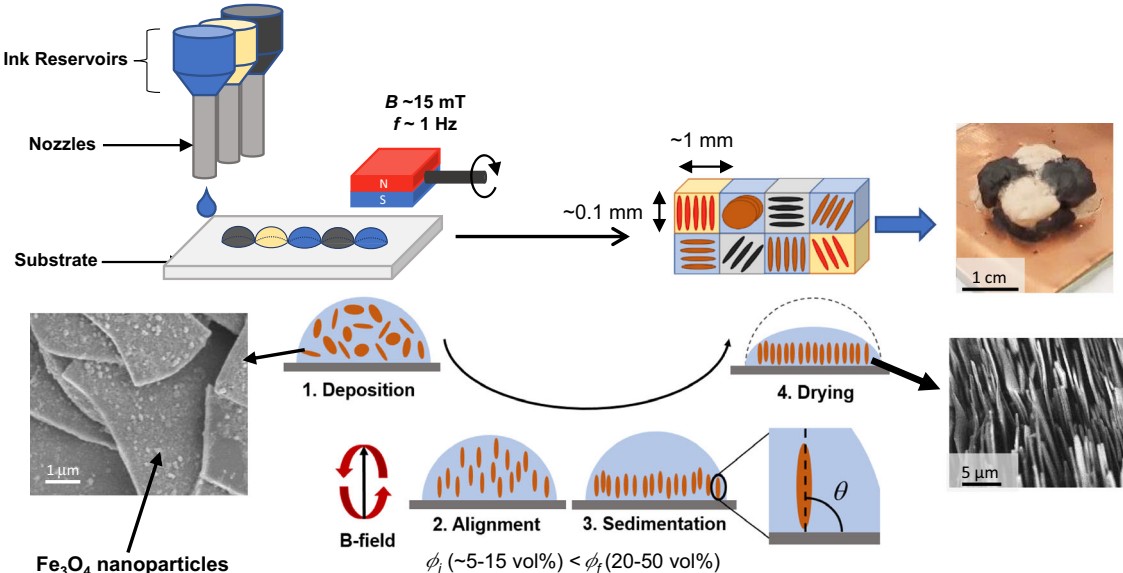

**Fig. 1 | Overview of magnetically assisted drop-on-demand printing (MDOD).** In MDOD, droplets of inks containing magnetically responsive microplatelets are deposited by a printer and a rotating magnetic field is applied to the printed droplets. The bottom left inset shows the electron micrograph of the microplatelets functionalised with superparamagnetic iron oxide nanoparticles to make them magnetically responsive. In summary, the droplets undergo four key steps. After the droplets are deposited on the substrate (Step 1. Deposition), the microplatelets are aligned by the magnetic field with field strength $B$ and rotational frequency $f$.

(Step 2. Alignment) while sedimentation occurs (Step 3. Sedimentation) until the solvent is completely dried (Step 4. Drying). The final platelet concentration $\phi_f$ is higher than the initial concentration $\phi_i$ during this process, leading to densified structures. The bottom right inset shows the electron micrograph of a dried, aligned droplet. The final printed structure can exhibit voxelated positional control of material and platelet orientation. Top right inset shows an example of a voxelated multimaterial and microstructured object produced by this technique. The black material is graphite and the light-coloured material is boron nitride.

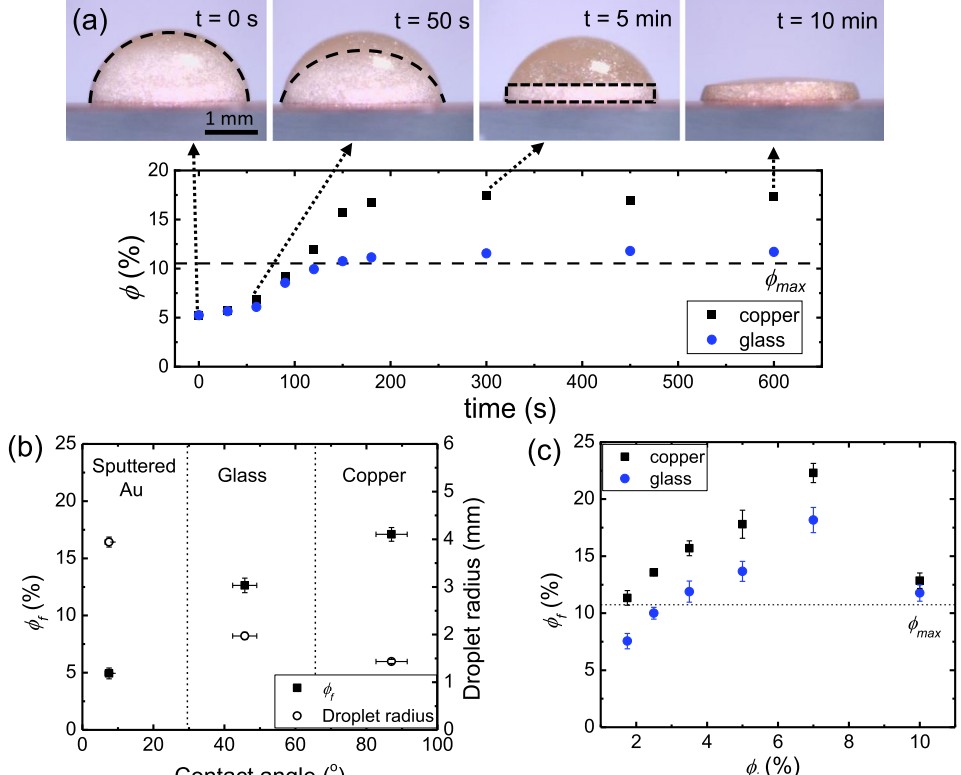

**Fig. 2 | Increase of platelet concentration in the deposited and vertically aligned xirallic droplets. a** Top panel shows the optical images of a vertically aligned xirallic droplet during sedimentation and drying on a copper substrate. The dotted region represents the volume which the xirallic platelets become concentrated during sedimentation. Bottom panel shows the platelet volume fraction $\phi$ as a function of time for droplets deposited onto two different substrates, copper (black) and glass (blue), for an initial ink concentration of 5 vol%. $\phi_{max}$ represents

the maximum volume fraction of microplatelets which can be dispersed in the ink solvent. **b** Variation of final platelet concentration, $\phi_f$ and droplet radius with contact angle of the droplet on three different substrates. **c** Relation between initial platelet concentration in the ink, $\phi_i$ and final platelet concentration, $\phi_f$ of the droplets drying on copper and glass substrates. All error bars represent the standard deviation of the measurements.

concentration. Although this causes stability issues for the ink stored in the 3D printer, this can be remedied by applying vibrations to the ink reservoirs to keep the ink dispersed prior to printing. Finally, once all the solvent has evaporated, a structure of aligned microplatelets held together by the polymeric binder remains. By continuously depositing inks with different material compositions, MDOD can fabricate multi-material voxelated structures. A polymer matrix can also be infiltrated into the printed structure to form a microstructured composite material. MDOD can achieve high degree of control as each droplet forms a voxel unit of the overall printed structure. In this study, we used water as the ink solvent due to its high surface tension to maximise the resolution of each voxel unit.

**Ink optimisation for orientation control and high concentration**
One key factor for attaining good control of microplatelet alignment and densification is the optimisation of the microplatelet concentration in the ink. We first used titania-coated alumina (xirallic) microplatelets to study the ink densification and alignment as they have a reflective platelet surface which allows easy identification of alignment during the printing process.

The drying behaviour of a single ink droplet was studied since it forms the basis of the 3D printing process. Due to their relatively large platelet dimensions, the maximum platelet content in the ink was approximately $\phi_{max} = 10.5$ vol% (Supplementary Fig. 1). Ink droplets with xirallic concentration of $\phi_i = 5$ vol% were deposited on two different substrates, copper and glass, subjected to vertical magnetic fields and the evolution of the droplet profiles were observed using an optical microscope (Fig. 2a). From the optical images, the shiny xirallic

platelets were initially dispersed within the droplet. Their shiny appearance indicated that they were vertically aligned to the magnetic field. After some time, they started to sediment due to their relatively large size and density as compared to water. By tracking the volume that the xirallic particles were dispersed in, the change in platelet concentration could be estimated (see Supplementary Information for details). The platelet concentration increased with time to a final value, $\phi_f$ of 17.5 vol% and 13 vol% for copper and glass substrates respectively. These values of $\phi_f$ were far beyond both the initial concentration $\phi_i$ and the maximum platelet content $\phi_{max}$ of 10.5 vol%. This is likely due to the magnetically induced alignment of the microplatelets with each other, which allows them to pack much closer together compared to a randomly aligned sample. Similar degrees of densification occurred regardless of the targeted microplatelet alignment. This is an exciting result as this opens the possibility to print structures with high platelet content which is not constrained by $\phi_{max}$. The presence of the magnetic field is imperative as droplets that are dried without an external magnetic field exhibited random microstructures and a non-flat top surface which makes them unfavourable for 3D printing (Supplementary Fig. 2).

The difference in $\phi_f$ observed between the copper and glass substrates is likely due to differences in contact angles of the ink on different substrates. This was verified by depositing ink droplets onto gold-sputtered glass substrates which has a lower contact angle than copper and glass (Supplementary Fig. 3). It is observed that $\phi_f$ increased with the contact angle, which is attributed to the degree of spreading of the droplet (Fig. 2b). Droplets with higher contact angles spread out over smaller areas. As a result, there are more

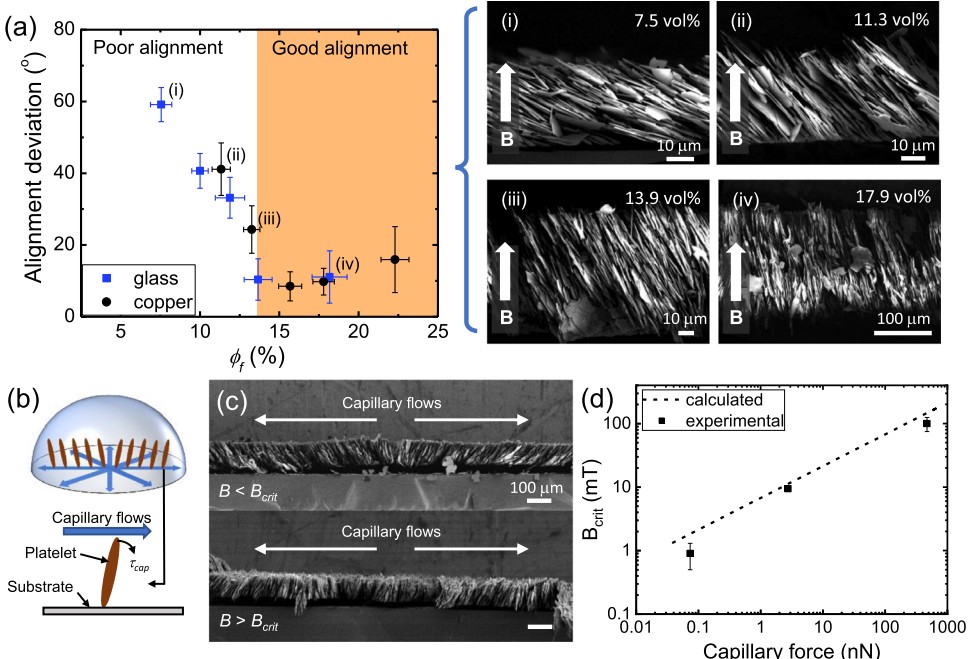

**Fig. 3 | Optimal conditions for effective magnetic alignment demonstrated with the xirallic ink. a** Variation of the microplatelet alignment deviation with respect to the target 90° angle in the dried droplet deposited on glass (blue) and copper (black) substrates as a function of the final platelet concentration $\phi_f$. The shaded area shows the region where the platelets are aligned according to the target 90° angle. (i), (ii), (iii) and (iv) refer to the images on the right which are cross-sectional electron micrographs of the droplets at $\phi_f$ = 7.5, 11.3, 13.9 and 17.9 vol%, respectively. **b** Schematics of the capillary flows generated within a drying sessile

droplet and the resulting torque $\tau_{cap}$, acting on the platelets within the droplet. **c** Micrographs of the cross-sections of droplets dried under a magnetic field of 7.5 mT, which is below the critical field strength $B_{crit}$ (top), and under a magnetic field 15 mT, which is above $B_{crit}$ (bottom). Both droplets had $\phi_i$ of 5 vol% and were deposited on a glass substrate. The target alignment was vertical (90°). **d** $B_{crit}$ as a function of the capillary forces experienced by a platelet. Experimental data points are fitted with calculated data (dotted line). All error bars represent the standard deviation of the measurements.

microplatelets per unit area and when they sediment, they exert a larger downward force on the lower layers of microplatelets which packs them closer together.

To identify the upper limit of $\phi_f$, we further tuned the initial microplatelet concentration $\phi_i$ of the ink (Fig. 2c). In general, an increase in $\phi_i$ led to an increase in $\phi_f$, until the initial platelet concentration became too large. The ink with $\phi_i$ = 10 vol%, which was close to $\phi_{max}$ showed minimal increase in $\phi_f$ after drying. In addition, the droplet profile shape for this ink remained spherical after drying similar to droplets that were not aligned with magnetic fields. This behaviour can be attributed to the high platelet concentration that resulted in high ink viscosity which impeded the rotation of the platelets during alignment. The lack of magnetic alignment in the droplet limited the densification as the platelets cannot attain a more orderly and packed structure. For the xirallic ink, the highest achievable $\phi_f$ was around 22.5 vol%, obtained for $\phi_i$ = 7.5 vol%.

Aside from contributing to a high platelet concentration in the dried printed droplet, a high $\phi_f$ is also needed for maintaining the alignment after drying. During drying, a capillary force arises between neighbouring microplatelets which cause them to collapse against each other and disrupts their alignment[20]. A high microplatelet concentration increases the resistance to this capillary force and keeps the microplatelets aligned towards the direction of the magnetic field.

To verify this phenomenon, the cross-sections of droplets with varying $\phi_f$ printed under a target alignment angle of $\theta$ = 90° were characterised (Fig. 3a). The deviation of the microplatelets' alignment from the target $\theta$ was plotted against $\phi_f$. As expected, the xirallic microplatelets could not retain the targeted 90° alignment at low $\phi_f$. The electron micrographs show that the microplatelets collapsed likely due to the capillary forces. As $\phi_f$ increased, the alignment of the microplatelets improved. When $\phi_f$ increased beyond 14 vol%, the xirallic platelets become well aligned to the target alignment with a

misalignment less than 18°, which is similar to what is obtained in other magnetically-oriented vertical structures[21]. It is interesting to note that this trend in platelet alignment is only dependent on $\phi_f$ regardless of the substrate used. This allows for flexible choice of substrates, as long as the initial microplatelet concentration is tuned to optimise $\phi_f$.

Another factor that helps maintain the final platelet alignment is the strength of the magnetic field applied. The magnetic field generates a torque on the microplatelet, which is usually opposed by viscous and gravitational torques[22]. In MDOD, the microplatelets experience an additional torque which misaligns them. In sessile droplets, capillary flows directed radially outward from the droplet centre develop during drying[23]. As these capillary flows move past the aligned microplatelet, they exert an additional torque that rotates the platelets towards the substrate (Fig. 3b). Therefore, a magnetic field larger than a critical magnetic field, $B_{crit}$ should be applied during printing for the microplatelets to maintain their target alignment (Fig. 3c). When the magnetic field is weaker than $B_{crit}$, the microplatelets tilt towards the outer edges of the droplet due to the capillary flows (Fig. 3c, top). Conversely, when the magnetic field applied is stronger than $B_{crit}$, the microplatelets remained vertically aligned throughout the droplet after drying (Fig. 3c, bottom).

To determine the minimum magnetic field strength required for good alignment, we estimated $B_{crit}$ by combining mathematical models related to capillary flows in sessile droplets and magnetic torques during magnetic alignment (see the Supplementary Information for details). In brief, we used a model by Deegan et al. to estimate the capillary flows within a droplet[24]. The capillary flows are fundamentally dependent on the contact angle of the droplet and the solvent evaporation rate. Since the capillary flows are time and position-dependent, an average value was computed for different droplets. The force and torque generated by the flow was then estimated. Lastly, this capillary torque was combined with other relevant torques in the

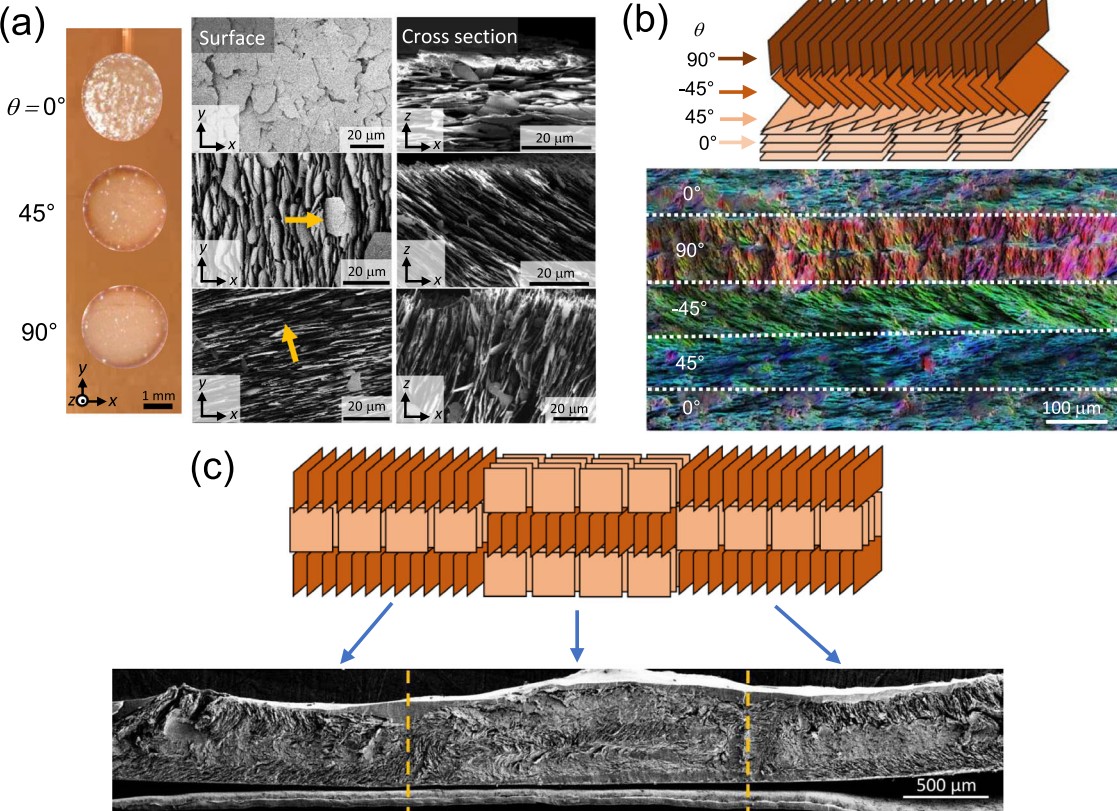

**Fig. 4 | Printing voxelated microstructures using the optimised xirallic ink on copper. a** Optical images of printed aligned droplets with microplatelet alignment at different angles $\theta$. The electron micrographs show the surface morphology (middle) and cross-section (right) of the corresponding droplets. The orange arrows on the surface morphology images represent the normal vector of the platelets which is indicative of their lateral directions. **b** Schematic of a multilayer structure with varying platelet angles $\theta$ in each layer and the electron micrograph of the cross-section of the corresponding printed structure after infiltration with a resin. The image is colour-coded based on the microplatelet alignment using OrientationJ. **c** Schematic of a 3 × 3 voxelated sample printed with varying platelet orientations in both horizontal and vertical directions and the micrograph of the corresponding printed structure after infiltration with a polymeric matrix. The vertical yellow dotted lines indicate the separation between the three columns of the design.

system, and $B_{crit}$ was estimated when the magnetic torque balances with the total opposing torques. The estimated values were then matched with experimental values, which were measured by varying the magnetic field applied to droplets deposited on different substrates (Fig. 3d). The mathematical model can thus be used to estimate the magnetic field strength that should be applied to the droplets during printing.

**Printing voxelated microstructured composites**

Having optimised the ink concentration to achieve controlled platelet alignment within single droplets, voxelated structures with localised control of microplatelet orientation were printed using MDOD (Fig. 4). Figure 4a shows different orientations in individual droplets, which form the basis of a voxel. The cross-sectional electron micrographs show that the microplatelet angle $\theta$ can be varied as desired from 0° to 90°. The lateral direction in which the microplatelets face can also be tuned as observed from the surface images.

Vertical micropillar structures can then be fabricated by depositing ink droplets onto each other during printing. The printing of droplets onto pre-existing printed parts should be done before the underlying structure is completely dried as the dried structures are porous and will create high capillary pressures that disrupt the alignment of the fresh ink (refer to Supplementary Information for more details). Since the platelet alignment in each droplet can be varied, the resulting layers within the micropillar can have varying platelet orientations. Figure 4b shows an example of one such structure which was printed with repeating microplatelet angles in a predesigned order

of 0°, 45°, −45° and 90°. In this sample, the as-printed structure was infiltrated with an epoxy to form a composite material. From the colour-coded micrograph of the cross-section, the microplatelet alignment was unperturbed by the infiltration and the varying alignment was apparent in each distinct layer. By printing these micropillar structures beside each other, fully voxelated structures can be achieved. This was demonstrated by printing a 3 × 3 unit voxelated structure with alternating 0° and 90° aligned microplatelet voxels (Fig. 4c). The voxels can be printed laterally or vertically as long as the pre-existing structure is not completely dried.

Closer inspection of the voxels in Fig. 4c revealed regions of misorientation ~150 μm wide at the boundaries between horizontally adjacent voxels (high magnification images in Supplementary Fig. 4). These regions are likely to be caused by the wetting behaviour between adjacent voxels. To verify this, we observed what happened when a fresh ink droplet was deposited next to an existing droplet (Supplementary Movie 1). From the recorded movie, it was clearly seen that there was some solvent flow from the fresh droplet towards the existing droplet. These flows are most likely Marangoni flows which exist due to the surface tension gradient between the droplets which is created by the different dispersant concentrations between a fresh droplet and a partially dried one[25]. In addition, these boundaries were less pronounced at ~50 μm, between adjacent voxels with same alignments (Supplementary Fig. 4c). This slightly mitigates the issue as most printed structures would not require a frequent change in platelet orientation laterally. Therefore, majority of the structure design (>85%) can still maintain good alignments as desired. In addition, this

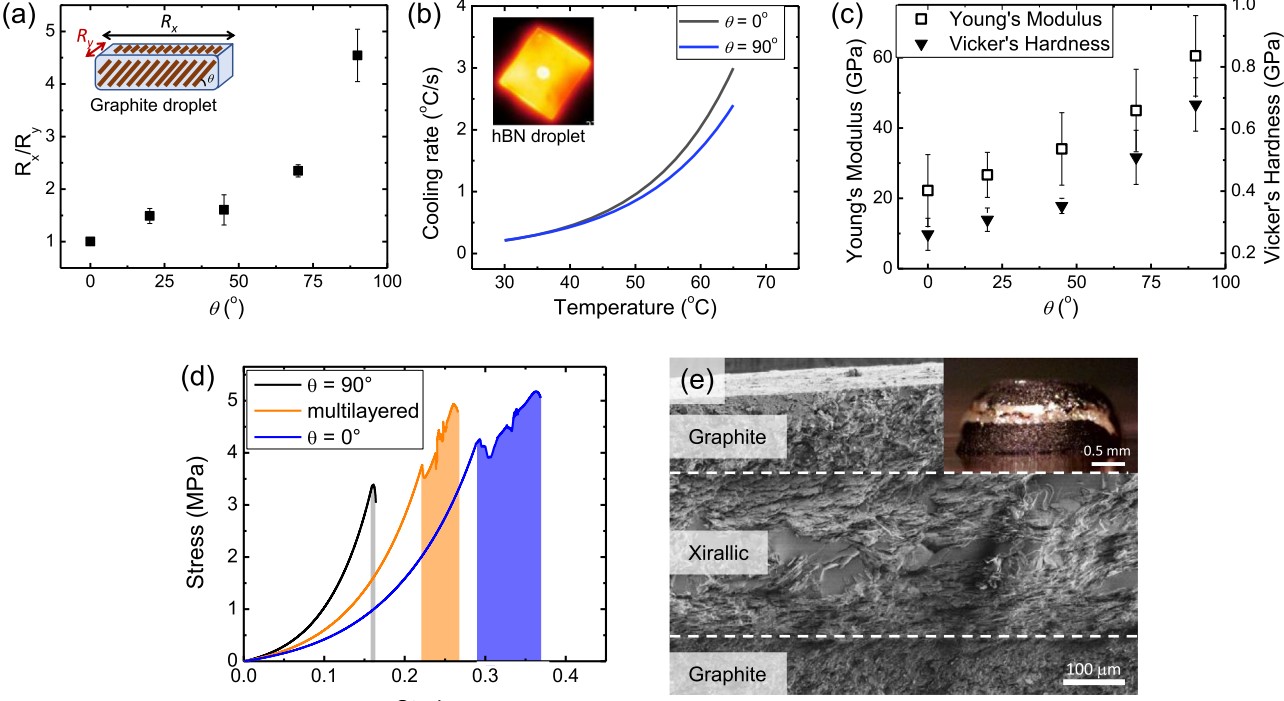

**Fig. 5 | Multimaterial printing and microstructure-dependent properties.**
**a** Variation of the electrical resistance anisotropy $R_x/R_y$ in graphite structures as a function of microplatelet angle $\theta$. Inset shows the directions at which $R_x$ and $R_y$ were measured relative to the platelet angle $\theta$. **b** Cooling rates of hBN droplets aligned at 0° (black) and 90° (blue). The insert shows a thermal image of a printed hBN droplet on a silicon wafer substrate, cooling down after being heated up to 70 °C. **c** Variation of the Young's modulus and Vicker's hardness of sintered xirallic droplets with the microplatelet angle $\theta$. **d** Representative stress–strain curves from compression testing performed on xirallic-PDMS structures printed with $\theta = 0°$ (purple), $\theta = 90°$ (black) and multi-layered (orange) alignments. Shaded area under graph represents energy dissipated from the first cracking event until failure. **e** Electron micrograph of the cross-section of a printed graphite-xirallic-graphite capacitor. The inset shows an optical image of the capacitor. All error bars represent the standard deviation of the measurements.

wetting behaviour has an advantage as it causes the droplet boundaries to fuse into a straight line and fill any gaps that would have existed between the two individual circular voxels.

## Multimaterial MDOD and microstructure-properties relationships

In addition to printing voxelated microstructures in highly concentrated composites, MDOD is applicable to a variety of microplatelets dimensions and materials. To demonstrate this, we printed aligned droplets using graphite, hBN and copper microplatelets which were made magnetically responsive as described earlier (Supplementary Fig. 5). Despite having different physical properties such as dimensions and densities, all microplatelets can be printed and magnetically aligned by MDOD after optimising the inks (Supplementary Table 1). Smaller microplatelets like graphite and hBN required a higher $\phi_i$ to achieve alignment and could attain a higher $\phi_f$ (-50 vol%) than what is currently reported in other methods. In addition, the anisotropic nature of these functional platelets leads to anisotropic properties that are tunable in the 3D printed structures (Fig. 5). By studying the relation between the microplatelet orientation angle and the material properties, a large design space is created to control the properties of printed structures.

First, tunable electrical properties were achieved by printing graphite droplets with varying $\theta$. The electrical resistances along the directions $x$ and $y$, denoted by $R_x$ and $R_y$ respectively, were measured and the ratio $R_x/R_y$ was tabulated against $\theta$ (Fig. 5a). It is observed that $R_x/R_y$ increased as the alignment of the graphite microplatelets was increased from 0° to 90°. This is due to the intrinsic anisotropy in each graphite microplatelet which has higher conductivity parallel to the platelet surface than through its thickness[26]. As the graphite orientation increased from $\theta = 0°$ to 90°, the relative degree of conduction

through the platelets thickness increased for $R_x$. Conversely, $R_y$ remained relatively constant regardless of $\theta$. Therefore, $R_x/R_y$ increased with $\theta$.

Next, tunable anisotropic thermal properties were obtained using printed hBN droplets. Here, hBN droplets with alignments at $\theta = 0°$ and 90° were printed on silicon substrates to emulate their use as thermal management material in electronics. The samples were heated to 70 °C, and their cooling rates back to room temperature were captured using a thermal imaging camera (Fig. 5b). The thermal conductivity is determined by percolation and hBN orientation. Similar to graphite, hBN has higher thermal conductivity along the surface of the microplatelet than through its thickness[27]. In addition, the percolation threshold is also lower when the hBN are aligned along the in-plane direction. Since the two different orientations have similar volume fractions of hBN microplatelets, the percolation should also favour thermal conduction in the in-plane direction as well. Therefore, it is expected that the sample with $\theta = 90°$ should cool faster by dissipating heat through the top surface of the droplet. However, the opposite trend was observed instead. This observation is most likely due to interfacial effects between the silicon and hBN. The vertically oriented hBN has a much smaller contact area with the heated silicon compared to the horizontally aligned hBN. In addition, while the horizontally aligned hBN has lower thermal conductivity towards the droplet surface, they have high conductivity towards the droplets side edges since that is the in-plane direction of the microplatelets. This would allow heat to be transferred quickly towards the edge of the printed structure to cool the sample.

Furthermore, tunable mechanical properties were demonstrated with the xirallic droplets. Here, xirallic structures printed and aligned at different $\theta$ were sintered at 1600 °C to form ceramic samples. SEM imaging was performed to verify that the microstructures of the

samples were as expected from the printing process (Supplementary Fig. 6b–d). Ceramics have stronger mechanical properties than polymer composites enabling us to better probe the variation of their properties with $\theta$. Both the Young's modulus and Vicker's hardness of the ceramics increased with $\theta$, from 22 GPa and 0.26 GPa respectively, for horizontally aligned samples to 60 GPa and 0.68 GPa respectively, for vertically aligned samples (Fig. 5c). These trends agree with reported studies in the literature which found that the microplatelets are mechanically stronger along the edge compared to the surface[28,29]. Therefore, our MDOD printing allows tunable mechanical stiffness solely through the control of the orientation angles of the microplatelets.

This design capability can be exploited to obtain a balance between properties that are difficult to achieve simultaneously such as stiffness, strength and toughness as hierarchical structures can help to promote energy dissipation though crack deflections[19,30]. We demonstrate this by printing a multi-layered xirallic-PDMS composites with alternating layers with microplatelet orientations of $\theta = 0°$ and 90° and samples with single alignments for comparison (Supplementary Fig. 6e). The samples were characterised using compression testing and their stress-strain curves were obtained (Fig. 5d). The $\theta = 90°$ sample had high stiffness of 4.7 MPa but had the lowest dissipated energy of 13.8 kJ m$^{-3}$. Conversely, the $\theta = 0°$ sample dissipated the most energy (361 kJ m$^{-3}$) but had a much lower stiffness of 2.17 MPa. By building alternating 0–90° aligned layers, the multi-alignment sample can achieve a compromise between stiffness (3.6 MPa) and energy dissipation (176 kJ m$^{-3}$). While the vertically aligned layers contribute to stiffness, the horizontal layers help prevent catastrophic failure by creating energy dissipating layers through mechanisms such as crack deflection (see Supplementary Fig. 6f)[31].

Finally, as a proof-of-concept example of multimaterial 3D printed structure, we printed graphite-xirallic-graphite layered structures and infiltrated them with epoxy to form a capacitor. The epoxy holds the structure together while having good dielectric strength which prevents dielectric breakdown during charging. The resultant structure forms a parallel plate capacitor with conductive graphite layers sandwiching a dielectric xirallic layer (Fig. 5e). The graphite layers were aligned at $\theta = 90°$ to ensure high conductivity between the dielectric layer and the electrical contacts. From the optical and electron micrographs, the graphite and xirallic layers were clearly distinguishable. The capacitance of our printed device (~3 mm diameter, <1 mm high) was in the range of 0.1 nF (see Supplementary Fig. 7 and Supplementary Discussion for details about the capacitor fabrication and performance).

## Printing multifunctional devices

To demonstrate the strengths of MDOD, multimaterial piezoresistive pressure sensors were fabricated. The multimaterial capability and control of microstructure of MDOD were leveraged to boost the printed device performances (Fig. 6). Piezoresistive pressure sensors are typically based on carbon filler-PDMS composite films that are often fabricated with microstructured surfaces such as micropillars and microdomes to increase their sensitivity[32,33]. MDOD can be used to create similar structures and incorporate additional filler alignment to further boost the sensor performances. In addition, hBN was incorporated into our printed sensor structures to create additional heat dissipation properties in these sensors which would be beneficial during device usage.

To fabricate the sensors, an array of micropillars of aligned graphite microplatelets was printed and infiltrated with PDMS (Fig. 6a). An additional ring of hBN was also deposited around these graphite micropillar for thermal management. The partial flow of fresh hBN ink into the micropillar ensured complete coverage and good interface between the two materials. The hBN and graphite microplatelet orientations were controlled independently using MDOD (Fig. 6b, c).

By sandwiching these micropillars between two copper tapes which act as the electrodes, a flexible piezoresistive sensor was formed. When pressure is applied onto the sensor area, the graphite fillers move closer to each other and the overall resistance of the sensor is reduced.

We first explore the sensor output of these printed sensors. The microstructural control in MDOD allows the mechanical properties of the sensors to be tuned (Fig. 6d). When $\theta = 90°$, the material is stiffer and therefore undergoes less strain than for $\theta = 0°$. This would affect the resultant sensing performance as the resistivity of the material is dependent on the strain experienced. The material was also highly elastic up to pressures of 1.5 MPa and it could return to its initial state as shown by the unloading curves. To test the electrical response to applied pressures, we applied small pressures onto the sensor area using a coin while monitoring the sensor resistance. The sensor resistance decreased when the coin was placed onto the sensor and returned to its original value when the coin was removed (Fig. 6e and Supplementary Movie 2). The consistent and repeatable change in resistance shows that the printed sensor can be used reliably. The sensor was further tested under a wider range of applied pressures by applying an increasing pressure was applied onto the sensor while monitoring its resistance (Supplementary Fig. 8 and Movie 3). The sensitivity, S of the sensor can be calculated using the following formula:

$$S = \frac{\triangle I/I_0}{\triangle P} \tag{1}$$

where $I_O$ is the default current through the sensor material when no pressure is applied and $\triangle I$ is the change in the current through the sensor when a pressure of $\triangle P$ is applied. Fig. 6f shows the measured electrical response of the sensors with different alignments. The sensitivity of the horizontally aligned samples was -0.91 kPa$^{-1}$, which is higher than the vertically aligned samples which had a sensitivity of 0.16 kPa$^{-1}$. This is expected given that the horizontally aligned samples are less stiff and experience a larger strain with applied pressure. The higher strain then corresponds to a larger change in resistance of the structure. While some recently reported flexible piezoresistive sensors have achieved higher sensitivities of -5–136 kPa$^{-1}$[34–36], they usually only work within a limited pressure range of <2.5 kPa. Conversely, our pressure sensor has a much wider sensing range up to 300 kPa, which covers a broader range of applications. The wide sensing range is only possible due to the high microplatelet content of the printed structures which gives it strong mechanical properties to withstand high pressures without being damaged.

In addition, the high degree of control in microplatelet orientation in MDOD can be leveraged to further tune the sensor output characteristics by combining layers with different graphite orientations. One additional desirable property in sensors is the linearity of the output signal. A large linear range is advantageous as signal analysis is simpler and require less complex circuits to operate[37]. We explore the use of MDOD to achieve this. While the $\theta = 0°$ samples have higher sensitivities, they exhibit linearity range of only up to 80 kPa. Conversely, the $\theta = 90°$ sample have lower sensitivities but a generally more linear response. Through simple empirical modelling using the measured stress–strain curve and piezoresistive characteristics, the overall sensor response for micropillars with varying ratio of $\theta = 0°$ and 90° layer thicknesses was estimated (details in Supplementary Information). From the modelling, we found that micropillars with 40% $\theta = 0°$ aligned graphite generated a combination of a large linear response range with good sensitivity. The results were then verified experimentally (Fig. 6f). Indeed, these multilayer alignment samples showed a wider linear response up to 300 kPa and with higher sensitivities than the $\theta = 90°$ samples.

By examining the modelling results, we also gained some insight on this observation. At low applied pressures, the detected signal is

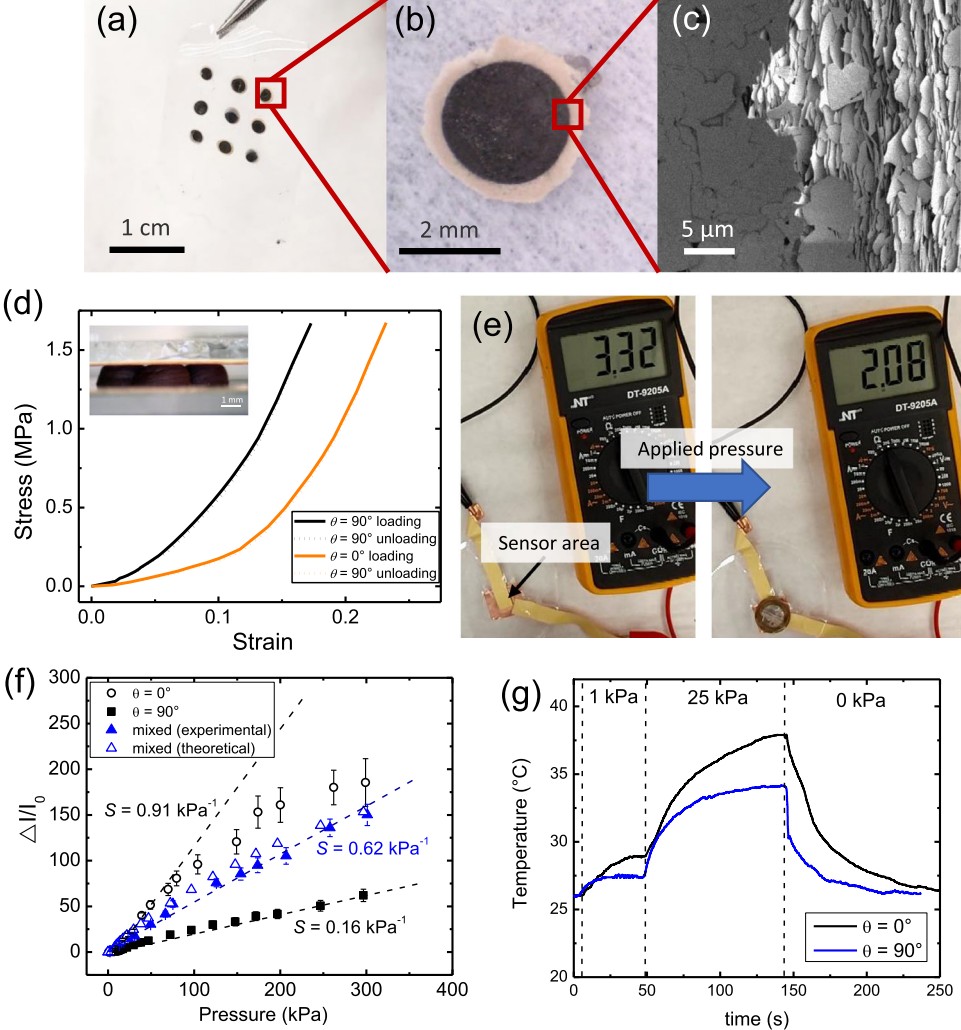

**Fig. 6 | Piezoresistive pressure sensor printed using MDOD. a** Optical image of MDOD printed piezoresistive pressure sensor consisting of an array of hBN-graphite micropillars embedded in a PDMS film. **b** Optical micrograph of one individual micropillar in the sensor. **c** SEM image of the surface of the micropillar showing control of orientation of the graphite (darker region) and hBN (lighter region). **d** Stress–strain curves of printed graphite-PDMS composite micropillars as piezoresistive pressure sensor with vertical and horizontal platelet alignment. Inset shows the optical image of the sensor structure whereby the micropillars are sandwiched between copper electrodes. **e** Photographs showing a reduction in resistance under the pressure exerted by the weight of a one-dollar coin. **f** Variation in detected electrical current in the sensor circuit with applied stress onto sensors with varying graphite alignments. $\Delta I/I_0$ is the fractional change in current when pressure is applied onto the sensor. $S$ is the sensitivity of the sensors, estimated by the initial gradient of the graph, which is highlighted by the dotted lines. Mixed alignment consists of 40% $\theta = 0°$ layers and is represented by the blue data points. **g** Temperature of the sensors with hBN orientations of $\theta = 0°$ (black) and $\theta = 90°$ (blue) under different applied pressures. All error bars represent the standard deviation of the measurements.

predominantly from the $\theta = 0°$ layer since it has a lower stiffness. However, the overall detected signal is reduced compared to a purely $\theta = 0°$ sample due to two factors. Firstly, the overall change in current, $\Delta I$ is reduced due to the presence of the less responsive $\theta = 90°$ layer. At the same time, the $\theta = 90°$ layer has a lower resistance, therefore, the default current $I_0$ is higher. This then corresponds to a lower signal of $\Delta I/I_0$. At higher pressures, the stiffer $\theta = 90°$ begins to experience higher strains and contribute to the sensor signals. These factors led to an overall larger linear signal characteristics. The ability to vary the relative number of layers with different microplatelet orientation makes MDOD a very powerful technique as a large spectrum of properties can be achieved depending on the needs of the final application.

Finally, the thermal properties of the printed sensors were studied. An issue with piezoresistive pressure sensors is the Joule heating effect when under operation[38]. As a current is constantly being passed through the sensor, heat is dissipated constantly depending on the resistance of the material. During practical applications, the heating tends to increase when pressure is applied since the sensor resistance decreases. This may lead to overheating or thermal drifts in the detected signal as changes in temperatures will also contribute to variations in resistance of the material. In addition, heating is also undesirable in certain applications such as wearable sensors to prevent burns to the user. Therefore, it is desirable to minimise device heating during operation. By manipulating the hBN orientation, we can maximise the heat dissipation as demonstrated earlier. To demonstrate this, we placed the sensors under a thermal camera while applying a force onto the sensor area to monitor the temperature fluctuations (Supplementary Fig. 9). Using weights, pressures of ~1 kPa and 25 kPa were applied onto the sensor and the equilibrium temperatures of the sensors under these pressures were measured (Fig. 6g).

In general, it was observed that the sample with hBN aligned at $\theta = 90°$, which is parallel to the micropillars, had a lower equilibrium temperature than samples with $\theta = 0°$. This is as expected since the alignment of $\theta = 90°$ enhanced the thermal conduction of heat towards the copper contacts which can efficiently conduct the heat away from

the device. Therefore, the device remains relatively cooler. Conversely, the $\theta = 0°$ alignment only facilitates thermal conduction towards the insulating air surrounding the micropillars and more heat is built up, leading to a higher equilibrium temperature. In addition, the heating and cooling also occurred more quickly in the $\theta = 90°$ sample, further verifying the increased thermal conduction away from the device. Further cyclical testing was performed to show the repeatability of these findings (Supplementary Fig. 9c). These results are in line with our earlier findings. The $\theta = 90°$ hBN provides both good thermal contact and fast thermal conduction in the lateral direction relative to the heated surface.

## Discussion

In addition to the functional capabilities of MDOD, the technique also allows for good control of printing resolution. Since the printed structure is made up of individual droplets, the lateral and vertical printing resolution is determined by the dimensions of the dried printed droplet. This is controlled by the contact angle of the ink on the substrate and the volume dispensed in each droplet. An increase in contact angle decreases the droplet diameter and increases the layer height. The volume of the droplet is tuned by changing the nozzle diameter and input pressure of the drop-on-demand printer. A larger droplet volume increases both the diameter and height of the droplet (Supplementary Fig. 10). Also, varying the droplet volume does not affect the magnetic alignment of the microplatelets in the droplets as the determining factor $\phi_f$ is independent of droplet volume (Supplementary Fig. 11). Overall, the lateral and vertical printing resolution achievable using the xirallic ink and copper substrates is ~0.7 mm and 50 μm, respectively. Comparatively, related printing methods reported by Martin et al. achieved resolutions of ~90 μm[15]. While the vertical resolution of MDOD exceeds that of related techniques, the lateral resolution is worse as it is limited by the minimum droplet volume dispensable by our printer. Nevertheless, the resolution can theoretically be improved to tens of microns by using printheads with smaller diameters or by incorporating high-resolution techniques such as electrohydrodynamic printing[39,40].

In terms of printing throughput, MDOD as a drop-on-demand technique that prints voxel by voxel will naturally be lacking compared to SLA or DIW, which can print entire layers of material quickly. Also, the use of water as the primary solvent in this study increases drying time for each droplet, which may make the process more arduous. To accelerate the printing speed, several strategies were explored.

Firstly, low heating of 50°C to the substrate decreased the drying time of each voxel by ~70% from ~10 min to ~2–4 min depending on the chosen droplet size (Supplementary Fig. 12). Only low heat was applied to curb the increase in capillary flows to ensure well-aligned structures could still be made. Further increase in heating would also be possible, but users should compensate the increase in capillary flows by applying stronger magnetic fields. Another strategy is to add volatile co-solvents into the ink. Using an ink with 25% ethanol and 75% water as the solvent, the drying time was further reduced by another 30%. The decrease in drying time arose from having a more volatile solvent and the reduction in contact angle which increased the surface area for evaporation. However, this inadvertently caused the printing resolution to decrease and therefore we limited the ethanol content to 25%. Higher ethanol content would still be possible if printing larger structures at higher throughput is a priority over resolution. Lastly, depending on the printed structure design, nearby voxels with the same microplatelet orientation can be printed and aligned concurrently as only a weak field is required for alignment (Supplementary Movie 4). The overall optimised processing time for our prints is listed in Table 1. The printing throughput could theoretically be further increased by improving the magnetic field source to cover a larger area on the print bed. This would allow multiple prints to be performed simultaneously.

**Table 1 | Typical processing times and dimensions for various samples**

| Sample | Shape | Dimensions | Optimised processing time |
|---|---|---|---|
| Checkered sample (Figure 1) | Cuboid | Length: 1.2 cm Breadth: 1.2 cm Height: 5.5 mm | ~6.67 hours |
| Multilayer sample (Fig. 4b) | Cylindrical | Diameter: 4 mm Height: 1.5 mm | ~30 min |
| Voxelated sample (Fig. 4c) | Cuboid | Length: 6 mm Breadth: 4 mm Height: 0.5 mm | ~10 min |
| Nanoindentation test sample (Fig. S6a) | Cuboid | Length: 8 mm Breadth: 8 mm Height: 1.5 mm | ~1.5 hours |
| Compression test sample (Fig. S6e) | Cylindrical | Diameter: 4 mm Height: 3 mm | ~1 hour |
| Pressure sensor (Fig. 6a) | 3 × 3 Array of cylindrical pillars | Length: 1.0 cm Breadth: 1.0 cm Height: 1.5 mm | ~45 min |

In conclusion, we developed MDOD as a technique to print multimaterial microstructured materials based on microplatelet orientation control. We studied the mechanism of MDOD and identified the microplatelet concentration and magnetic field strength as key parameters to attain precise filler alignment during printing. Using this knowledge, we applied MDOD to inks with different materials to achieve voxelated control of localised material and microstructure. Overall, MDOD offers three key advantages. Firstly, the final printed structures can achieve high solid content in the range of 20–50 vol% while allowing good control of the microplatelet alignment. Next, the microstructural control gives MDOD the versatility to achieve a large tunability in the printed material properties. Therefore, a good balance between multiple properties that are otherwise difficult to achieve can be easily done in MDOD. Lastly, MDOD is compatible with multimaterial printing to incorporate multiple functionalities into the printed structures. These capabilities were demonstrated by fabricating sensitive piezoresistive pressure sensors which demonstrated high linear sensing range with added thermal management properties.

While this study focuses the understanding of MDOD as a technique, we envision that its potential can be further realised by applying MDOD with functional anisotropic nanoscale materials with superior properties such as MXenes and other 2D nanomaterials to fabricate devices with exceeding performances[41,42]. Such nanomaterials have been increasingly used with 3D printing to fabricate structures for energy and electronics applications[43,44]. Using MDOD, we can provide an additional factor of microstructuring to these printed devices to improve device performances. In addition, MDOD can also be extended to print reinforced composites with 1D materials since they can be easily aligned with static magnetic fields[19,45]. Overall, the versatility of MDOD to exert control of the microstructure, utilise different materials and compositions can create a large design space to fabricate a wide range of microstructured composites with tunable properties.

## Methods
### Materials

Xirallic titania-coated alumina microplatelets (Merck, average diameter ~20 μm, thickness ~200 nm), polyvinylpyrrolidone MW 360,000 (Sigma-Aldrich), superparamagnetic iron oxide nanoparticles ferrofluid (Ferrotec EMG-605), hBN microplatelets (Merck, average diameter ~10 μm, thickness ~300 nm), graphite (Merck, average diameter ~7 μm, thickness ~300 nm), PDMS (SYLGARD™ 184) and epoxy resin (Weicon MS1000) were purchased and used without modification.

## Ink preparation

The different microplatelets were first made magnetically responsive by adsorbing SPIONs on their surfaces. In a typical procedure, 2 g of dry microplatelets were first dispersed in 200 ml of deionized (DI) water using magnetic stirring. Ferrofluid EMG-605 was then added to the suspension such that the SPIONs made up 5 vol% to the relative mass of the platelets. The mixture was stirred overnight and filtered using vacuum filtration to recover the magnetised microplatelets. The microplatelets were then dried in a drying oven overnight. To make the ink, the dried magnetised platelets were mixed with 1 wt% PVP aqueous solution at the desired microplatelet content. The resulting mixture was then sonicated and vortexed until a homogenous ink was produced.

## Ink properties

The ink rheology was characterised using a shear rheometer (Bohlin Gemini HR Nano). Viscometry measurements were made using a 15 mm diameter serrated plate system with 200 μm gap size and shear rates from $0.1 s^{-1}$ to $500 s^{-1}$. Each measurement was repeated three times. The ink contact angle was characterised by taking side view optical images of ink droplets ~6 μL using a USB microscope (Dino-lite AM7915MZTL). The contact angles of five ink droplets were measured for each type of ink. Sedimentation rates were measured by taking a time-lapse of ink droplets ~6 μL until they totally dry.

## 3D printing

3D printing was performed using an automated fluid dispensing system (Nordson 3-Axis PROPlus). The ink was filled into 5 ml syringes (Nordson) before loading into the printer. A 0.33 mm inner diameter stainless steel flat tip needle (Nordson) was then installed onto the printhead. The input air pressure was set at ~0.5 bar. The copper substrates were prepared by attaching copper foil (Sunhayato) onto glass slides (VWR) to keep them flat. Glass coverslips (VWR) were used as glass substrates. The gold-sputtered substrates were prepared by sputtering (Joel JFC-1600) the glass coverslips with ~10 nm of gold. The substrates were cleaned with ethanol prior to printing to ensure consistent printing as impurities on the substrates may lead to a variation in ink contact angle and could cause defects in the print.

The printing steps of the desired pattern were programmed onto the printer's software interface. A rotating magnet consisting of a permanent Neodymium magnet attached to a DC motor (RS Components) was set up at the side of the printer. During each printing step, several droplets were deposited on the substrate and the stage was programmed to move the droplets to the rotating magnet set up for alignment (Supplementary Movie 4).

The printed structures were infiltrated with a PDMS or epoxy matrix to form composites. The resins and the hardeners were pre-mixed according to the specifications from the manufacturer and degassed in vacuum for ~10 mins. An appropriate amount of the degassed matrix material was then deposited onto the printed structures and the samples were placed under a vacuum for ~1 hour for infiltration. The samples were then kept in an oven at 40 °C for 1 day to completely cure the matrices. After curing, any excess matrix material can be removed by mechanically filing the samples.

## Characterisation of printed samples

The cross-sectional areas of the printed samples were characterised using field-effect scanning electron microscopy (JOEL 6340 F) and ImageJ analysis to identify the alignment of the printed microplatelet composites. The angles of at least 50 microplatelets were taken on each sample.

## Electrical resistance measurements

Graphite droplets with varying alignment of $\theta = 0°$, 20°, 45°, 70°, 90° and diameter of ~4 mm and heights of ~0.2 mm were printed on glass substrates. Three samples of each alignment were printed. 4 thin strips of carbon tape ~1 mm × 3 mm were attached to opposite edges of each droplet along the $x$ and $y$ directions as defined in Fig. 5a. The carbon tapes acted as electrical contacts for the resistance measurements. Electrical resistance measurements were then made using a two-probe multimeter (NT DT-9205A). Five measurements were taken for each sample.

## hBN cooling rate

hBN droplets with platelet alignments of $\theta = 0°$ and 90° and diameters of ~4 mm were deposited onto the middle of a $1.8 \times 1.8 cm^2$ silicon wafer (Merck). After drying, the samples were placed in a convection oven (IKA 125-control) and preheated at 80 °C for 10 mins. The samples were then removed and placed onto an insulating cardboard under a thermal camera (FLIR ETS320). A movie of the sample cooling down was recorded. The cooling rate was then obtained from the slope of the temperature vs time profile of each sample. The measurements were performed at least three times for each sample.

## Mechanical properties of xirallic structures

Four by four xirallic droplet structures with varying alignments were printed with an overall length and breadth of 8 mm and height of 1.5 mm. Once the droplets were dried, they were sintered in a high-temperature furnace (Nabertherm LHT 08/18) first at 500 °C for 1 hour for binder removal and then at 1600 °C for 2 hours for sintering. Once the samples were cooled back to room temperature, they were cold mounted in an epoxy resin for subsequent preparation for further characterisation. The mounted samples were first grinded using sandpapers with increasing grits of 400, 800, 1200 and 2400. This was followed by polishing using OPS solution (Struers). The mechanical properties of the polished samples were characterised using nanoindentation (G200, KLA Tencor, US) and Vicker's hardness tester (Future Tech FM-300E). The nanoindentation tests were performed using a Berkovich tip with a loading rate of $1 mN s^{-1}$ to a maximum load of 100 mN and dwell time of 10 s. 20 indents were made on each sample. For the Vicker's hardness test, a load of 1 kg was applied for 10 s. Nine indents were made on each sample.

Xirallic-PDMS structures were also fabricated for toughness measurements using compression tests. Xirallic micropillars with diameters of 4 mm and height of 3 mm were printed and infiltrated with PDMS. The samples were then subjected to compression testing (Instron 3366) using a 500 N load cell and a loading rate of $0.2 mm min^{-1}$. Three samples of each orientation were tested for repeatability.

## Piezoresistive pressure sensor fabrication and characterisation

Three by three arrays of graphite droplets were printed into micropillars with diameters of ~2.0 mm and heights of ~1.5 mm on a copper foil substrate to form a sensor area of ~1 cm by 1 cm. Rings of hBN were then deposited around each micropillar and the structure was infiltrated with PDMS to form the final device. The stress–strain curve and electromechanical properties of the resultant sensors were tested using a compression tester (Instron 3366). The two pieces of copper foils sandwiching the printed graphite micropillars were connected a two-probe multimeter. The electrical resistance was monitored while a compressive force with a ramp rate of 5 N min⁻¹, up to a maximum of 40 N was applied to the sensor. To check the repeatability and stability of the sensor signals, the measurements were performed after every 10 compression cycles up to 30 cycles.

The thermal properties of the sensors were characterised using a thermal camera (FLIR ETS320). A lever system using a tweezer and a support was used to apply a force onto the sensor area. This was done to ensure that the sensor area would be unobstructed for the camera

to record its temperature. Weights were placed onto the middle of the tweezer to apply pressures equivalent to 1 kPa and 25 kPa onto the sensor area. For each loading and unloading step, the temperature was recorded until a saturation temperature was reached.

## Data availability

The data that support the conclusions presented herein are available from the authors upon request. Supplementary Information is provided with this paper.

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

## Acknowledgements
We acknowledge the Facility for Analysis, Characterisation, Testing and Simulation (FACTS), Nanyang Technological University, Singapore, for the use of their scanning electron microscopy facilities. This research was funded by the National Research Foundation of Singapore (Award NRFF12-2020-0002, H.L.F).

## Author contributions
W.C.L. and H.L.F. conceived the idea. W.C.L. performed the experiments with support from VHYC and R.P.B. W.C.L. analysed and modelled the data. All authors discussed the results and wrote the paper.

## Competing interests
The authors declare no competing interests.
