## [Peer Review File · Nature Communications]

Title: Magnetically assisted drop-on-demand 3D printing of microstructured multimaterial compositesREVIEWER COMMENTS

Reviewer #1 (Remarks to the Author):

In this manuscript, the authors present a droplet-based assembly process of anisotropic particles under applied magnetic fields. These droplets evaporate upon deposition and leave behind highly oriented and quite dense microstructures of particles. By using polymeric solutions in the precursor fluid, the authors can arrive at composite materials with drop-wise programmability of the microstructure. The authors display various functional outcomes including mechanical and electrical. This reviewer's overall assessment is that the authors have provided a neat droplet evaporation approach that produces nice looking structures, but the authors haven't sufficiently analyzed the properties of these structures nor have they sufficiently developed the performance of any "devices" made from this manufacturing approach to justify publication in an impact journal. In its current state, this reviewer does not recommend acceptance. If the authors can add additional rigor to the characterization of properties and device performance, and if the authors more accurately describe the realities of the state of their manufacturing process, then this reviewer would be happy to change this recommendation.

1. Reading in between the lines, it appears this process is quite arduous suggesting the authors may be using too rosy of language to treat this as a next generation manufacturing process. Each droplet takes 10 minutes to evaporate, and during that evaporation time, other droplets likely can't be deposited since application of different magnetic fields would disrupt the alignment. Thus, this process can create 6 "voxels" per hour at best. Yet the authors indicate this is a new manufacturing process. This reviewer encourages the authors to use blunt language to describe the state of the approach (i.e. arduous) to the readers.

2. Figure 4a and 4b demonstrate very well ordered microstructures for isolated droplets and droplets deposited in a vertical pattern, respectively. Figure 4c reveals that when droplets are placed more in a 3d grid (as the suggested manufacturing approach is intended) the microstructure outcome shows significant inconsistencies. There appear to be large undulations in the structure (insufficiently discussed by authors). The image is also of the appropriate resolution so as not to allow visual analysis of microstructural alignment, especially at the boundary between "voxels". This reviewer requests that the authors analyze the boundary between horizontally adjacent voxels and discuss the challenges or opportunities with their findings. Does a horizontally adjacent voxel "wet" a neighboring voxel and create capillary flow that will disrupt alignment during the evaporation process? This reviewer imagines that would be the case and would create a limitation.

3. On the topic of functional electrical devices with oriented graphene in Fig 5a. This reviewer would highlight that the differences between a conductive material and a resistive material are 12 orders of magnitude in resistance. The authors are reporting a difference less than 1 order of magnitude in electrical conductivity. This anisotropy therefore doesn't seem significant enough for this to be enable new applications. Magnetic alignment for enhanced conductivity (thermal and electrical), has been widely reported, so this reviewer is looking for novelty here.

4. On the topic of functional thermal devices with oriented hBN in Fig 5b. The authors find a different trend than expected but don't spend additional effort to validate or understand the trend. This reviewer would suggest that thermal conductivity is dominated by percolation and orientation. The authors have no discussion of percolation. Further, the surface of a very good insulator would cool before the interior; perhaps the oriented hBN is in fact allowing more heat to escape. Whether true or not, overall this topic seems underthought.

5. On the topic of functional mechanics the authors report mechanics from sintered ceramics derived from the droplet evaporation study. The ability to generate a bulk coupon that can be tested in a universal tester such as an Instron feels like a compelling result. This reviewer would like the authors to include images of the test coupons along with microstructural analysis to prove the structure was as expected. These results would anyway defend impact of this work in an ability to manufacture bulk 3d parts. If instead the authors somehow devised a mechanical test to characterize very small samples (single droplet or a few droplets large), then the authors need to explain their approach and defend the rigor in a method that is not to testing standards.

6. On the topic of toughness and resiliency in Fig 5d. The authors show crack deflection as proof of toughness. Crack deflection does not always mean a material is tougher. If the alignment of the platelets in this case create an easy-to-cleave path through a material, the material could in fact require much less energy to fracture. To demonstrate toughness, some measurement of the actual energy required to fracture is needed to establish that there is microstructure-enhanced toughness.

7. On the topic of the capacitor in Fig 6. The authors demonstrate a multi-material capacitor that uses conductive flake composites sandwiching a dielectric flake composite. The capacitor works and is sensitive to pressure. This type of capacitor could be manufactured with conventional processes in a scalable way, and with more optimized materials. It is not clear the benefit that these specific materials bring over current capacitors, nor is it clear what benefit the microstructure itself provides relative to the library of current capacitor materials. This reviewer would ask the authors to better defend why this is a landmark demonstration of this manufacturing approach as would be expected in a high impact journal.

8. There is no demonstration of an advantageous property or performance metric with the multiple orientations possible in a printed structure. This feels required to prove the arduous manufacturing process is a meaningful scientific or technological contribution.

Reviewer #2 (Remarks to the Author):

Review of Magnetically assisted drop-on-demand 3D printing of microstructured multimaterial composites

In this paper, the authors proposed a novel additive manufacturing technique able to locally control the orientation of platelets (of various types) incorporated into the ink. There has been a lot of work in the recent years to create materials with a texture (local orientation of platelets). Many processes have been reported to control the short-range order. Alternatively, many processes already exist to control the shape of the pieces. This study starts filling an interesting gap, with the demonstration of both short-range order (locally aligned platelets) and medium-range order (different domains with different platelet orientations within each domain). There are plenty of natural materials with such (or similar) complex microstructures, but achieving this level of control in synthetic materials has proved challenging.

The structural and functional benefits of such control are demonstrated with several examples and demonstration devices. The study is well conceived, well executed and clearly explained. The authors build upon two well-developed processes (magnetically assisted slip casting and 3D printing). The results are thus novel, convincing, and useful for further developments of the materials. Not all problems associated to this novel approach are solved, but the mechanisms controlling the microstructure are identified and explained. I assume the work will continue in this group to expand the technology, but there is already plenty in this paper. This study is a nice step towards a more elaborated control of the microstructure of materials and thus certainly of interest to the materials science community and beyond.

Specific comments:

- Figure 1, inset on the right: there are apparently two different materials printed here (black and white), although this is not explained in the caption, please specify
- line 145-160: the alignment and drying stages are well explored. According to the authors, there is a competition between settling and the alignment induced by the magnetic field, when the field exceeds an identified threshold. Isn't gravitation sufficient to align the platelets during settling? Do they settle parallel or perpendicular to the surface? Did the authors tried to investigate the drying of such systems without applying a magnetic field?
- Is it possible that a pre-alignment takes place during the droplet deposition, because of the shear applied upon the deposition of droplets, or is the fraction of platelets in the ink too low for this?
- Figure 2 (and corresponding text): how is the parameter ϕ (volume fraction of platelets) measured? By extracting the apparent volume of droplets?
- Figure 3: since there is a radial flow during drying from the center towards the edge, are there conditions where there is a central defect with less platelets once drying is completed?
- Figure 3a: the authors defined two domains: « poor alignment » and « good alignment ». Is the threshold arbitrary? How was it defined/decided? There is apparently a minima, with the disorientation increasing again for a higher platelet concentration.
- Drying: the authors mentioned (line 286-287) that any additional layer should be deposited before the previous layer is « completely dried », which is a bit qualitative. Is the environment controlled to ensure a constant humidity? Is there any parameter that could be used to control this step in a reproducible way, or does it really depend on the experience of the user?
- Figure 4: This is a very nice demonstration of the fine control of orientation achieved. It could be worth

color-coding the SEM imaged in 4b, for example with OrientationJ in ImageJ

<http://bigwww.epfl.ch/demo/orientation/>

- Figure 4: there seems to be variations of thickness in the final samples. Any idea of their origin? Is the shrinkage different depending on the platelet orientation? Is it due to a wetting of the additional droplet that depend on the platelet orientation in the layer below? What is the maximum number of layers the authors printed? Comments (if any) on the shape control of the final pieces would be welcome.

- Figure 4: the blue dotted lines are not very visible, maybe try yellow?

- Figure 5d: orientation based color-coding would be helpful here as well to identify the approximate location of the boundary between successive layers.

- methods: is the cleaning of the glass substrates critical for the process? sometimes it can impact the wetting properties, which seems to be critical in this case (Fig 2b)

- indications about the typical processing times and dimensions of the pieces would be nice to add to the paper

- sensors (last part of the paper): although the results seem convincing, I have no particular expertise on sensors so I cannot really comment on this. I nevertheless appreciate the authors efforts to go all the way to demonstration devices.

Please find below our answers to the reviewer's comments. Changes made to the main manuscript are in green.

Reviewer #1 (Remarks to the Author):

In this manuscript, the authors present a droplet-based assembly process of anisotropic particles under applied magnetic fields. These droplets evaporate upon deposition and leave behind highly oriented and quite dense microstructures of particles. By using polymeric solutions in the precursor fluid, the authors can arrive at composite materials with drop-wise programmability of the microstructure. The authors display various functional outcomes including mechanical and electrical. This reviewer's overall assessment is that the authors have provided a neat droplet evaporation approach that produces nice looking structures, but the authors haven't sufficiently analyzed the properties of these structures nor have they sufficiently developed the performance of any "devices" made from this manufacturing approach to justify publication in an impact journal. In its current state, this reviewer does not recommend acceptance. If the authors can add additional rigor to the characterization of properties and device performance, and if the authors more accurately describe the realities of the state of their manufacturing process, then this reviewer would be happy to change this recommendation.

Response: First and foremost, we would like to thank you for the time taken in reviewing our manuscript and for providing detailed comments. Second, we would like to further emphasize that the aim of this study is to develop a 3D printing method that allows fine tuning of the microstructure. To do so, we employed common and easily available microplatelets to quantify and demonstrate the capabilities of the MDOD printing presented. Employing more advanced and highly performing microplatelets in the composition (for example graphene instead of graphite) or matrix (for example by grafting the polymeric matrix to the alumina microplatelets) would therefore be expected to further increase the performance of the printed parts.

In the following, we address each comment in detail and put efforts in detailing the capabilities and limitations of the proposed method.

1. Reading in between the lines, it appears this process is quite arduous suggesting the authors may be using too rosy of language to treat this as a next generation manufacturing process. Each droplet takes 10 minutes to evaporate, and during that evaporation time, other droplets likely can't be deposited since application of different magnetic fields would disrupt the alignment. Thus, this process can create 6

"voxels" per hour at best. Yet the authors indicate this is a new manufacturing process. This reviewer encourages the authors to use blunt language to describe the state of the approach (i.e. arduous) to the readers.

Response: Thank you for your comments. We agree that the localized control MDOD gives does come at the expense of the overall process throughput. This is due to the drop-on-demand nature and the use of water as the main ink solvent. We have updated the manuscript to describe this more accurately for readers.

In addition, we also studied and discuss some methods to speed up the process, which include applying mild heating to the substrate, using a volatile co-solvent and printing droplets in parallel. We have also now included additional data related to the approximate processing times of droplets of different sizes to let readers have an accurate gauge of the time frames involved. While the overall printing speed is not comparable with methods such as stereolithography or direct ink writing, we believe that the MDOD has advantages over those techniques in terms of local composition and microstructural control, making MDOD a more effective method for applications that can make use of these strengths, such as for electronic components.

Edits to manuscript:

In the discussion section:

In terms of printing throughput, MDOD as a drop-on-demand technique that prints voxel by voxel will naturally be lacking compared to SLA or DIW, which can print entire layers of material quickly. Also, the use of water as the primary solvent in this study increases drying time for each droplet, which may make the process more arduous. To accelerate the printing speed, several strategies were explored.

Firstly, low heating of 50 °C to the substrate decreased the drying time of each voxel by approximately 70 % from ~10 min to ~2-4 min depending on the chosen droplet size (**Supplementary Figure S12**). Only low heat was applied to curb the increase in capillary flows to ensure well-aligned structures could still be made. Further increase in heating would also be possible, but users should compensate the increase in capillary flows by applying stronger magnetic fields. Another strategy is to add volatile co-solvents into the ink. Using an ink with 25 % ethanol and 75 % water as the solvent, the drying time was further reduced by another 30 %. The

decrease in drying time arose from having a more volatile solvent and the reduction in contact angle which increased the surface area for evaporation. However, this inadvertently caused the printing resolution to decrease and therefore we limited the ethanol content to 25%. Higher ethanol content would still be possible if printing larger structures at higher throughput is a priority over resolution. Lastly, depending on the printed structure design, nearby voxels with the same microplatelet orientation can be printed and aligned concurrently as only a weak field is required for alignment (**Supplementary Video S4**). The overall optimised processing time for our prints is listed in **Supplementary Table S2**.

Addition to Supplementary Information:

Figure S12: Droplet drying time. Time taken for droplets to dry under ambient conditions without heating, and with additional substrate heating at 50 °C.

Table S2: Typical processing times and dimensions for various samples

Sample	Shape	Dimensions	Optimised Processing Time
Checkered Sample (Fig. 1)	Cuboid	Length: 1.2 cm Breadth: 1.2 cm Height: 5.5 mm	~6.67 hours
Multilayer Sample (Fig. 4b)	Cylindrical	Diameter: 4 mm Height: 1.5 mm	~30 min
Voxelated Sample (Fig. 4c)	Cuboid	Length: 6 mm Breadth: 4 mm	~10 min

		Height: 0.5 mm	
Nanoindentation test sample (Fig. S6a)	Cuboid	Length: 8 mm Breadth: 8 mm Height: 1.5 mm	~1.5 hours
Compression test sample (Fig. S6e)	Cylindrical	Diameter: 4 mm Height: 3 mm	~1 hour
Pressure Sensor	3 x 3 Array of cylindrical pillars	Length: 1.0 cm Breadth: 1.0 cm Height: 1.5 mm	~45 min

2. Figure 4a and 4b demonstrate very well ordered microstructures for isolated droplets and droplets deposited in a vertical pattern, respectively. Figure 4c reveals that when droplets are placed more in a 3d grid (as the suggested manufacturing approach is intended) the microstructure outcome shows significant inconsistencies. There appear to be large undulations in the structure (insufficiently discussed by authors). The image is also of the appropriate resolution so as not to allow visual analysis of microstructural alignment, especially at the boundary between "voxels". This reviewer requests that the authors analyze the boundary between horizontally adjacent voxels and discuss the challenges or opportunities with their findings. Does a horizontally adjacent voxel "wet" a neighboring voxel and create capillary flow that will disrupt alignment during the evaporation process? This reviewer imagines that would be the case and would create a limitation.

Response: Thank you for the comment. Yes, there appears to be a region of misalignment between horizontally adjacent voxels. We have investigated this more closely by observing droplets deposited next to each other and we did observe a flow of solvent from the fresh droplet into the existing droplet. We have added a discussion about this in the manuscript, a video showing this flow and additional high magnification SEM images into the supplementary information. Overall, we find that a small region of misalignment is created between adjacent voxels, with an approximate width of $\sim 150 \mu\text{m}$ between voxels with different alignments and $\sim 50 \mu\text{m}$ between voxels with similar alignments. We estimate that the overall defective volume based on this effect would not exceed 15% based on a typical 2 mm x 2 mm voxel surrounded on all 4 sides with voxels with different alignments. This percentage will be much lower in practice as most voxels would be part of a bigger region with similar platelet alignments. In addition, we find that the voxel interface becomes straight instead of the curved circular droplet shape, which helps in filling any gaps that would have occurred in combining purely circular voxels next to each other.

Addition to Printing voxelated microstructured composites section:

Closer inspection of the voxels in **Fig. 4c** revealed regions of misorientation $\sim 150\ \mu\text{m}$ wide at the boundaries between horizontally adjacent voxels (high magnification images in **Supplementary Figure S4**). These regions are likely to be caused by the wetting behaviour between adjacent droplets. To verify this, we observed what happened when a fresh ink droplet was deposited next to an existing droplet (**Supplementary Video S1**). From the recorded video, it was clearly seen that there was some solvent flow from the fresh droplet towards the existing droplet. These flows are most likely Marangoni flows which exist due to the surface tension gradient between the droplets which is created by the different dispersant concentrations between a fresh droplet and a partially dried one.²⁵ In addition, these boundaries were less pronounced $\sim 50\ \mu\text{m}$, between adjacent voxels with same alignments (**Supplementary Figure S4c**). This slightly mitigates the issue as most printed structures would not require a frequent change in platelet orientation laterally. Therefore, majority of the structure design ($> 85\%$) can still maintain good alignments as desired. In addition, this wetting behaviour has an advantage as it causes the droplet boundaries to fuse into a straight line and fill any gaps that would have existed between the 2 individual circular voxels.

Figure S4: Boundary between horizontally adjacent voxels. (a) and (b) High magnification SEM images of the boundary between voxels in the sample presented in Figure 4c in the manuscript. The adjacent voxels have different microplatelet orientations. (c) Boundary between adjacent voxels with the same microplatelet alignments.

3. On the topic of functional electrical devices with oriented graphene in Fig 5a. This reviewer would highlight that the differences between a conductive material and a resistive material are 12 orders of magnitude in resistance. The authors are reporting a difference less than 1 order of magnitude in electrical conductivity. This anisotropy therefore doesn't seem significant enough for this to be enable new applications. Magnetic alignment for enhanced conductivity (thermal and electrical), has been widely reported, so this reviewer is looking for novelty here.

Response: Thank you for your comment. Indeed, the anisotropy presented is not sufficiently high to create any novel applications on its own. However, we think that this section mainly acts as an additional verification that when the platelet orientation is varied during MDOD, the resultant properties do vary accordingly. Although we are using microplatelets like graphite, our results suggest that employing high performing microplatelets would result in a larger enhancement of the properties. We mention this in our manuscript with:

Finally, while the structures in our study were fabricated using 2D microplatelets, **the concepts of ink formulation and magnetic response studied could be applied** to use MDOD with other functional anisotropic nanoscale materials such as MXenes and other 2D nanomaterials to fabricate devices with exceeding performances.^{40,41}

The brief discussion in the section is more to set up our work especially in the printed sensor in the subsequent section which is the more important and novel concepts that we would like to show in MDOD.

4. On the topic of functional thermal devices with oriented hBN in Fig 5b. The authors find a different trend the expected but don't spend additional effort to validate or understand the trend. This reviewer would suggest that thermal conductivity is dominated by percolation and orientation. The authors have no discussion of percolation. Further, the surface of a very good insulator would cool before the interior; perhaps the oriented hBN is in fact allowing more heat to escape. Whether true or not, overall this topic seems underthought.

Response: Thank you for your comment. We have added some discussion regarding the observations. Both the platelet orientation and percolation contribute to higher thermal conductivity along the in-plane direction of the hBN platelets. This is because the percolation threshold of the platelets is lower when they are aligned in the in-plane direction. Therefore, the vertically aligned hBN should have high conductivity perpendicular to the substrate and the horizontally aligned hBN should have high conductivity parallel to the substrate. We expected a faster cooling rate by the vertically aligned hBN by

dissipating heat from the top surface of the droplet. However, this was not the case. Therefore, we believe that this is due to interfacial effects that impede heat transfer between the substrate and the hBN droplet.

Additions to manuscript under Multimaterial MDOD and microstructure-properties relationships section:

Next, tunable anisotropic thermal properties were obtained using printed hBN droplets. Here, hBN droplets with alignments at $\theta = 0^\circ$ and 90° were printed on silicon substrates to emulate their use as thermal management material in electronics. The samples were heated to 70°C , and their cooling rates back to room temperature were captured using a thermal imaging camera (**Fig. 5b**). The thermal conductivity is determined by percolation and hBN orientation. Similar to graphite, hBN has higher thermal conductivity along the surface of the microplatelet than through its thickness.²⁶ In addition, the percolation threshold is also lower in along the in-plane direction of the microplatelet. Since the two different orientations have similar volume fraction of hBN microplatelets, the percolation should also favour thermal conduction in the in-plane direction as well. Therefore, it is expected that the sample with $\theta = 90^\circ$ should cool faster by dissipating heat through the top surface of the droplet. However, the opposite trend was observed instead. This observation is most likely due to interfacial effects between the silicon and hBN. The vertically oriented hBN has a much smaller contact area with the heated silicon compared to the horizontally aligned hBN. In addition, while the horizontally aligned hBN has lower thermal conductivity towards the droplet surface, they have high conductivity towards the droplets side edges since that is the in-plane direction of the microplatelets. This would allow heat to be transferred quickly towards the edge of the printed structure to cool the sample.

5. On the topic of functional mechanics the authors report mechanics from sintered ceramics derived from the droplet evaporation study. The ability to generate a bulk coupon that can be tested in a universal tester such as an Instron feels like a compelling result. This reviewer would like the authors to include images of the test coupons along with microstructural analysis to prove the structure was as expected. These results would anyway defend impact of this work in an ability to manufacture bulk 3d parts. If instead the authors somehow devised a mechanical test to characterize very small samples (single droplet or a few droplets large), then the authors need to explain their approach and defend the rigor in a method that is not to testing standards.

Response: We have included images of a typical sample which we used in our mechanical testing. The samples are typically 2 by 2 droplet samples with final dimensions of approximately $8 \times 8 \times 1.5 \text{ mm}^3$ with controllable platelet orientation in each droplet. We verify that after sintering, the microstructure in the sample is as determined by the initial microplatelet orientation as shown by the SEM images.

We performed mechanical testing on these samples using nanoindentation as this technique is able to give us localized modulus on the sample and allows us to correlate the microstructure with these properties. Nanoindentation is a well-studied technique that are commonly used to characterize materials with hierarchical structure such as those fabricated with MDOD. An example of the use of this technique can be found in:

Amini S, Tadayon M, Idapalapati S, Miserez A. The role of quasi-plasticity in the extreme contact damage tolerance of the stomatopod dactyl club. *Nat Mater.* **14**, 943-50 (2015).

Addition to supplementary figures:

Figure S6: Sample fabricated for mechanical testing. (a) Optical image of MDOD printed sintered xirallic sample used for mechanical testing. The different shades correspond to different microplatelet orientation. The lighter portion corresponds to $\theta = 0^\circ$ (horizontally aligned) and the darker portion corresponds to $\theta = 90^\circ$ (vertically aligned). (b) Microstructure of sample at the boundary to show the different microplatelet orientations. (c) and (d) High magnification SEM images of the microstructure in different microplatelet orientations. (e) Optical image of the sample. (f) High magnification SEM image of the microstructure showing a crack (indicated by a yellow arrow).

each region, showing that the microstructure is as expected. (e) Optical image of xirallic-PDMS composite for compression test. (f) Electron micrograph showing a crack deflection event in a multilayer xirallic-epoxy composite with varying alignments. The colour code corresponds to microplatelets with varying alignment angles.

Addition to manuscript Multimaterial MDOD and microstructure-properties relationships section

Furthermore, tunable mechanical properties were demonstrated with the xirallic droplets. Here, xirallic structures printed and aligned at different θ were sintered at 1600 °C to form ceramic samples. SEM imaging was performed to verify that the microstructures of the samples were as expected from the printing process (**Supplementary Figure S6b-d**).

This design capability can be exploited to obtain a balance between properties which are difficult to achieve simultaneously such as stiffness, strength and toughness as hierarchical structures can help to promote energy dissipation through crack deflections.^{19,30} We demonstrate this by printing a multilayered xirallic-PDMS composites with alternating layers with microplatelet orientations of $\theta = 0^\circ$ and 90° and samples with single alignments for comparison (**Supplementary Figure S6e**).

6. On the topic of toughness and resiliency in Fig 5d. The authors show crack deflection as proof of toughness. Crack deflection does not always mean a material is tougher. If the alignment of the platelets in this case create an easy-to-cleave path through a material, the material could in fact require much less energy to fracture. To demonstrate toughness, some measurement of the actual energy required to fracture is needed to establish that there is microstructure-enhanced toughness.

Response: We have performed energy dissipation measurements by using compression testing on MDOD printed xirallic-PDMS composite samples with alternating vertically ($\theta = 90^\circ$) and horizontally ($\theta = 0^\circ$) aligned layers. We have included an image of what a typical sample would look like in the SI. We also performed the measurements for purely vertical and horizontally aligned samples as comparison. The measurement is based on existing work in our group from the following reference:

Chan, X. Y., Chua, C., Tan, S. & Le Ferrand, H. Energy dissipation in composites with hybrid nacre-like helicoidal microstructures. *Compos. Part B Eng.* **232**, 109608 (2022).

Compression testing was performed as it can measure mechanical properties of smaller samples reliably to give fair comparison between the samples. As our objective to using the multilayer alignment is to induce crack deflection, we wanted to analyse the energy dissipated from such events. From the stress-strain curves, we obtained the energy dissipated during crack initiation and propagation by calculating the area under the graph at kinked portions. These regions are typically caused by initiation of cracks and their propagation within the structure which relieved the stress within the structure which causes the drop in stress in the sample.

We have slightly modified our message here, which is that MDOD allows for tuning of mechanical properties to obtain a balance between stiffness and toughness. We have added the discussion in the manuscript.

Addition to manuscript Multimaterial MDOD and microstructure-properties relationships section

This design capability can be exploited to obtain a balance between properties which are difficult to achieve simultaneously such as stiffness, strength and toughness as hierarchical structures can help to promote energy dissipation through crack deflections.^{19,30} We demonstrate this by printing a multilayered xirallic-PDMS composites with alternating layers with microplatelet orientations of $\theta = 0^\circ$ and 90° and samples with single alignments for comparison (**Supplementary Figure S6e**). The samples were characterised using compression testing and the stress strain curves were characterised (**Fig. 5d**). The $\theta = 90^\circ$ sample had high stiffness of 4.7 MPa but had the lowest dissipated energy of 13.8 kJm^{-3} . Conversely, the $\theta = 0^\circ$ sample dissipated the most energy (361 kJm^{-3}) but had a much lower stiffness of 2.17 MPa. By building alternating $0\text{-}90^\circ$ aligned layers, the multi-alignment sample can achieve a compromise between stiffness (3.6 MPa) and energy dissipation (176 kJm^{-3}). While the vertically aligned layers contribute to stiffness, the horizontal layers create energy dissipating layers by inducing crack deflection as reflected in the kinks in the compression curves (see **Supplementary Fig. S6f,g**).³¹

Figure 5: Multimaterial printing and microstructure-dependent properties. (a) Variation of the electrical resistance anisotropy R_x/R_y in graphite structures as a function of microplatelet angle θ . Inset shows the directions at which R_x and R_y were measured relative to the platelet angle θ . (b) Cooling rates of hBN droplets aligned at 0 and 90°. The insert shows a thermal image of a printed hBN droplet on a silicon wafer substrate, cooling down after being heated up to 70 °C. (c) Variation of the Young's modulus and Vicker's hardness of sintered xirallic droplets with the microplatelet angle θ . (d) Representative stress-strain curves from compression testing performed on xirallic-PDMS structures printed with varying microplatelet alignments. Shaded area under graph represents energy dissipated until failure. (e) Electron micrograph of the cross-section of a printed graphite-xirallic-graphite capacitor. The inset shows an optical image of the capacitor.

7. On the topic of the capacitor in Fig 6. The authors demonstrate a multi-material capacitor that uses conductive flake composites sandwiching a dielectric flake composite. The capacitor works and is sensitive to pressure. This type of capacitor could be manufactured with conventional processes in a scalable way, and with more optimized materials. It is not clear the benefit that these specific materials bring over current capacitors, nor is it clear what benefit the microstructure itself provides relative to the library of current capacitor materials. This reviewer would ask the authors to better defend why this is a landmark demonstration of this manufacturing approach as would be expected in a high impact journal.

Response: Thank you for the comment. We would first like to clarify that the capacitors and pressure sensors are two different demonstrations of MDOD printed devices. The original purpose of the capacitor is to showcase the multimaterial compatibility and printing capabilities of MDOD in fabricating functional

devices. Nevertheless, we agree that the performance of the capacitors is lacking as it would benefit from a more optimal material choice for both the electrodes and the dielectric material. Therefore, for this manuscript which we want to focus more on the strengths of MDOD as a processing technique, we have decided to remove the details about the capacitor performance to the Supplementary Materials and focus on the more in-depth properties and functionalities in the printed piezoresistive sensors which can better showcase the capabilities of MDOD.

We have updated the work on the piezoresistive sensor to explore more device performance properties, as well as a better understanding of how the microstructural control is contributing to performance of the sensors. This would be elaborated in comment 8 below.

Overall, the sensor application showcases three key advantages of MDOD as a processing technique as compared to other ink-based printing methods:

- (i) MDOD can achieve high solid loading composite materials. This directly contributes to the sensors mechanical robustness which translates to its large sensing range which is uncommon for flexible sensors.
- (ii) MDOD can achieve good microstructural control, with only 18 degree misorientation in vertically-aligned composites. We use this to achieve a good balance between the sensitivity and linearity of the sensor signal. Different combinations of platelet orientations can be exploited to create different mechanical and electrical properties in the composite material which interplay to give the final performance in the sensors. This is a powerful tool which gives a large tunability in the properties of the printed device based on what is required for the final application.
- (iii) MDOD is compatible with multimaterial printing. This is demonstrated by incorporating hBN as thermal management material to the sensors.

These 3 key advantages were added to the Discussion section:

Overall, MDOD offers three key advantages. Firstly, the final printed structures can achieve high solid content in the range of 20 -50 vol% while allowing good control of the microplatelet alignment with less than 18° misalignment in vertically oriented structures. Next, the microstructural control gives MDOD the versatility to achieve a large tunability in the printed material properties. Therefore, a good balance between multiple properties that is otherwise difficult to achieve can be easily done in MDOD. Lastly, MDOD is compatible with multimaterial printing to incorporate multiple functionalities into the printed structures.

8. There is no demonstration of an advantageous property or performance metric with the multiple orientations possible in a printed structure. This feels required to prove the arduous manufacturing process is a meaningful scientific or technological contribution.

Response: Thank you for your comment. We have now further developed our piezoresistive sensor application. We added 2 key results in this section. Firstly, we demonstrated advantageous properties in the sensor output when combining different graphite orientation in a single micropillar. We found that we can vary the relative composition of $\theta = 0^\circ$ and $\theta = 90^\circ$ within the micropillar to achieve sensors with a larger linear output range. We support this finding with empirical modelling from the stress-strain curves and electromechanical response measured from the $\theta = 0^\circ$ and $\theta = 90^\circ$ samples presented earlier. The second addition is the inclusion of additional hBN into the sensor to increase heat dissipation. The orientation of hBN in this case can affect how effective the heat dissipation is. Overall, we feel that the addition of these results highlights the key merits of MDOD by making use of varying platelet orientation and multimaterial properties as explained in comment 7 above. We believe that these additions will make this a landmark demonstration for MDOD that is suitable for a high impact journal.

Additions to the beginning in the “Printing multifunctional devices” section:

The overall strengths of MDOD makes it suited to fabricate structures that requires a high degree of localized control with precision, such as components and circuits for flexible electronics. To demonstrate this, multimaterial piezoresistive pressure sensors were fabricated. The multimaterial capability and control of microstructure of MDOD was leveraged to boost the printed device performances (**Fig. 6**). Piezoresistive pressure sensors are typically based on carbon filler-PDMS composite films that are often fabricated with microstructured surfaces such as micropillars and microdomes to increase their sensitivity.^{32,33} MDOD can be used to create similar structures and incorporate additional filler alignment to further boost the sensor performances. In addition, hBN was incorporated into our printed sensor structures to create additional heat dissipation properties in these sensors which would be beneficial during device usage.

To fabricate the sensors, an array of micropillars of aligned graphite microplatelets was printed and infiltrated with PDMS (**Fig. 6a**). An additional ring of hBN was also deposited around these graphite micropillar for thermal management. The partial flow of fresh hBN ink into the micropillar ensured complete coverage and good interface between the two materials. The hBN

and graphite microplatelet orientations were controlled independently using MDOD (Fig. 6b and c). By sandwiching these micropillars between two copper tapes which act as the electrodes, a flexible piezoresistive sensor was formed. When pressure is applied onto the sensor area, the graphite fillers move closer to each other, and the overall resistance of the sensor is reduced.

Figure 6: Piezoresistive pressure sensor printed using MDOD. (a) Optical image of MDOD printed piezoresistive pressure sensor consisting of an array of hBN-graphite micropillars embedded in a PDMS film. (b) Optical micrograph of one individual micropillar in the sensor. (c) SEM image of the surface of the micropillar showing control of orientation of the graphite (darker region) and hBN (lighter region). (d) Stress-strain curves of printed graphite-PDMS composite micropillars as piezoresistive pressure sensor with vertical and horizontal platelet alignment. Inset shows the optical image of the sensor structure whereby the micropillars are sandwiched between copper electrodes. (e) Photographs showing a reduction in resistance under the pressure exerted by the weight of a one-dollar coin. (f) Variation in detected electrical current in the sensor circuit with applied stress onto sensors with varying graphite alignments. $\Delta I/I_0$ is the

fractional change in current when pressure is applied onto the sensor. S is the sensitivity of the sensors, estimated by the initial gradient of the graph, which is highlighted by the dotted lines. Mixed alignment consists of 40% $\theta = 0^\circ$ layers. (g) Temperature of the sensors with different hBN orientations under different applied pressures.

Additions to the ending in the “Printing multifunctional devices” section:

In addition, the high degree of control in microplatelet orientation in MDOD can be leveraged to further tune the sensor output characteristics by combining layers with different graphite orientations. One additional desirable property in sensors is the linearity of the output signal. A large linear range is advantageous as signal analysis is simpler and require less complex circuits to operate.³⁷ We explore the use of MDOD to achieve this. While the $\theta = 0^\circ$ samples have higher sensitivities, they exhibit linearity range of only up to 80 kPa. Conversely, the $\theta = 90^\circ$ sample have lower sensitivities but a generally more linear response. Through simple empirical modelling using the measured stress-strain curve and piezoresistive characteristics, the overall sensor response for micropillars with varying ratio between $\theta = 0^\circ$ and 90° layer thicknesses was estimated (details in **Supplementary Information**). From the modelling, we found that micropillars with 40% $\theta = 0^\circ$ aligned graphite generated a combination of a large linear response range with good sensitivity. The results were then verified experimentally (**Fig. 6f**). Indeed, these multilayer alignment samples showed a wider linear response up to 300 kPa and with higher sensitivities than the $\theta = 0^\circ$ samples.

By examining the modelling results, we also gained some insight on this observation. At low applied pressures, the detected signal is predominantly from the $\theta = 0^\circ$ layer since it has a lower stiffness. However, the overall detected signal is reduced compared to a purely $\theta = 0^\circ$ sample due to two factors. Firstly, the overall change in current, ΔI is reduced due to the presence of the less responsive $\theta = 90^\circ$ layer. At the same time, the $\theta = 90^\circ$ layer has a lower resistance, therefore, the default current I_0 is higher. This then corresponds to a lower signal of $\Delta I / I_0$. At higher pressures, the stiffer $\theta = 90^\circ$ begins to experience higher strains and contribute to the sensor signals. These factors led to an overall larger linear signal characteristics. The ability of vary the relative amounts of layers with different microplatelet orientation makes MDOD a very powerful technique as a large spectrum of properties can be achieved depending on the needs of the final application.

Finally, the thermal properties of the printed sensors were studied. An issue with piezoresistive pressure sensors is the Joule heating effect when under operation.³⁸ As a current is constantly being passed through the sensor, heat is dissipated constantly depending on the resistance of the material. During practical applications, the heating tends to increase when pressure is applied since the sensor resistance decreases. This may lead to overheating or thermal drifts in the detected signal as changes in temperatures will also contribute to variations in resistance of the material. In addition, heating is also undesirable in certain applications such as wearable sensors to prevent burns to the user. Therefore, it is desirable to minimise device heating during operation. By manipulating the hBN orientation, we can maximise the heat dissipation as demonstrated earlier. To demonstrate this, we placed the sensors under a thermal camera while applying a force onto the sensor area to monitor the temperature fluctuations (**Supplementary Figure S9**). Using weights, pressures of approximately 1 kPa and 25 kPa were applied onto the sensor and the equilibrium temperatures of the sensors under these pressures were measured (**Fig. 6g**).

In general, it was observed that the sample with hBN aligned at $\theta = 90^\circ$, which is parallel to the micropillars, had a lower equilibrium temperature than samples with $\theta = 0^\circ$. This is as expected since the alignment of $\theta = 90^\circ$ enhanced the thermal conduction of heat towards the copper contacts which can efficiently conduct the heat away from the device. Therefore, the device remains relatively cooler. Conversely, the $\theta = 0^\circ$ alignment only facilitates thermal conduction towards the insulating air surrounding the micropillars and more heat is built up, leading to a higher equilibrium temperature. In addition, the heating and cooling also occurred more quickly in the $\theta = 90^\circ$ sample, further verifying the increased thermal conduction away from the device. Further cyclical testing was performed to show the repeatability of these findings (**Supplementary Figure S9c**). These results are in line with our earlier findings. The $\theta = 90^\circ$ hBN provides both good thermal contact and fast thermal conduction in the lateral direction relative to the heated surface.

Figure S9: Setup for thermal characterization of graphite sensor. (a) Setup used to study the Joule heating effect when pressure is applied onto the sensor. (b) thermal camera image showing the increase in temperature when a pressure of 25 kPa is applied onto the sensor. (c) Temperature of sensor under several cycles of loading and unloading of 25 kPa load.

Reviewer #2 (Remarks to the Author):

In this paper, the authors proposed a novel additive manufacturing technique able to locally control the orientation of platelets (of various types) incorporated into the ink. There has been a lot of work in the recent years to create materials with a texture (local orientation of platelets). Many processes have been reported to control the short-range order. Alternatively, many processes already exist to control the shape of the pieces. This study starts filling an interesting gap, with the demonstration of both short-range order (locally aligned platelets) and medium-range order (different domains with different platelet orientations within each domain). There are plenty of natural materials with such (or similar) complex microstructures, but achieving this level of control in synthetic materials has proved challenging.

The structural and functional benefits of such control are demonstrated with several examples and demonstration devices. The study is well conceived, well executed and clearly explained. The authors build upon two well-developed processes (magnetically assisted slip casting and 3D printing). The results are thus novel, convincing, and useful for further developments of the materials. Not all problems associated to this novel approach are solved, but the mechanisms controlling the microstructure are identified and explained. I assume the work will continue in this group to expand the technology, but there is already plenty in this paper. This study is a nice step towards a more elaborated control of the microstructure of materials and thus certainly of interest to the materials science community and beyond.

Specific comments:

1. Figure 1, inset on the right: there are apparently two different materials printed here (black and white), although this is not explained in the caption, please specify

Response: Thank you for the comment, the black material is graphite and white material is hBN. We have added this information into the caption of Fig. 1:

The black material is graphite and the light coloured material is boron nitride.

2. line 145-160: the alignment and drying stages are well explored. According to the authors, there is a competition between settling and the alignment induced by the magnetic field, when the field exceeds an identified threshold. Isn't gravitation sufficient to align the platelets during settling? Do they settle

parallel or perpendicular to the surface? Did the authors tried to investigate the drying of such systems without applying a magnetic field?

Response: Thank you for the comment. We did study the microplatelet behaviour without the application of magnetic fields. The microplatelets could not attain a well-aligned structure (see Figure below). The microplatelets near the bottom were generally horizontally aligned but the bulk of it was randomly aligned. Theoretically, gravity should align the platelets parallel to the substrate, but we hypothesize that the capillary flows which develops during drying may disrupt this leading to randomly aligned region. Also, it was noticed that the droplets obtained without the magnetic field did not have a flat substrate, which would make a 3D construction challenging (Figure S3 in SI).

We therefore added the following image to the Supplementary Information.

Figure S2: Effect of magnetic field on droplet shape and microstructure during drying. Optical images of the profile of a xirallic droplet (a) aligned with magnetic field and (b) dried without magnetic field. (c) SEM cross section of the microstructure of a xirallic droplet dried without magnetic field.

3. Is it possible that a pre-alignment takes place during the droplet deposition, because of the shear applied upon the deposition of droplets, or is the fraction of platelets in the ink too low for this?

Response: We believe that pre-alignment does not occur as the fraction of platelets in the ink is low. In addition, the low volume fraction also means the ink has very low viscosity and yield stress. Therefore, even if any pre-alignment did occur, it would not be maintained for long after the droplet is completely deposited on the substrate.

- Figure 2 (and corresponding text): how is the parameter phi (volume fraction of platelets) measured?
By extracting the apparent volume of droplets?

Response: Yes, the evolution of ϕ during the drying process is estimated by extracting the apparent volume of the droplets during drying. By conservation of mass, we can estimate ϕ using the following calculation:

$$\phi_i V_i = \phi_t V_t$$

Since we control the initial microplatelet volume fraction ϕ_i and the initial droplet volume V_i , we can obtain the microplatelet concentration at any time t , ϕ_t , by estimating the volume that the microplatelets occupy within the droplet at time t from the optical micrographs. This volume is outlined by the black dotted lines in the upper panel of Figure 2a. During settling, the volume is approximated as a spherical cap and when the platelets are fully settled, the volume is approximated as a cylinder. These details were added to the Supplementary Information.

- Figure 3: since there is a radial flow during drying from the center towards the edge, are there conditions where there is a central defect with less platelets once drying is completed?

Response: For the range of microplatelet concentrations studied, this scenario did not occur. We believe that this would occur at much lower microplatelet concentrations. However, at such low concentrations, alignment would not be possible and is therefore not of importance for the technique.

- Figure 3a: the authors defined two domains: « poor alignment » and « good alignment ». Is the threshold arbitrary? How was it defined/decided? There is apparently a minima, with the disorientation increasing again for a higher platelet concentration.

Response: For this part of the study, we set the threshold for “good alignment” as within 20% deviation from the target alignment angle, which is equivalent to 18° here. From our observation, part of the observed increase in misorientation at higher platelet concentrations may be due to damage induced when breaking the droplets for SEM examination. When breaking the samples, a compressive force is inadvertently applied to the samples. For lower concentration samples, this was not an issue. However, for higher platelet concentration and un-infiltrated samples, they seem to undergo slight buckling (see Fig. a below). This may have caused us to observe slight misalignment. However, when we infiltrated the samples with a polymer, this buckling behaviour was no longer observed (see Fig. b below). Therefore, it is reasonable to believe that good alignment is still achieved in high platelet concentration samples. In

any case, the slight increase in misalignment of the samples still fall within the range which we consider “good alignment”.

Figure: SEM images of cross section of (a) un-infiltrated xirallic droplets and (b) infiltrated xirallic droplets.

We added the following detail to the manuscript under section “Ink optimisation for orientation control and high concentration” after Figure 3:

When ϕ_f increased beyond 14 vol%, the xirallic platelets become well aligned to the target alignment with a misalignment of about 18° , which is similar to what is obtained in other magnetically-oriented vertical structures.²¹

- Drying: the authors mentioned (line 286-287) that any additional layer should be deposited before the previous layer is « completely dried », which is a bit qualitative. Is the environment controlled to ensure a constant humidity? Is there any parameter that could be used to control this step in a reproducible way, or does it really depends on the experience of the user?

Response: We performed our printing in ambient conditions without deliberately controlling the surrounding conditions. From our experience with this technique, the additional layer can be deposited any time after the platelets have sedimented as they will get jammed and will not be affected by subsequent magnetic field applied to the next droplet. To make this process more systematic and reproducible, we suggest that the next drop to be deposited when the previous droplet is approximately 80% dried. We have added a section in the Supplementary Information to describe this.

Addition to Supplementary Discussion:

Time interval between deposition of droplets during printing

To ensure good alignment in each voxel, the next droplet should be deposited on the existing structure before the underlying layer is fully dried. To make this process more systematic and reproducible, we recommend that a fresh droplet to be deposited on the existing structure based on a fixed time interval equivalent to 80% of the drying time of a droplet. The drying time of each droplet, t_{dry} should first be measured either by monitoring the side profile of the droplet using a microscope or by monitoring the mass of the droplet on a weighing balance if such a microscopy set up is unavailable. The overall drying time depends on the volume of each droplet and the environment of the user's laboratory (**Fig. S12**). The print steps can then be set to deposit one drop after a time interval, $t_{interval} = 0.8 t_{dry}$. We found that this time interval ensures a consistent and continuous process. At shorter $t_{interval}$, there will be an eventual accumulation of solvent in the structure which will cause the fresh ink to spill out from the existing structure and deteriorate the print resolution. At longer $t_{interval}$, the structure becomes close to being completely dried which may affect the alignment of the next droplet.

- Figure 4: This is a very nice demonstration of the fine control of orientation achieved. It could be worth color-coding the SEM imaged in 4b, for example with OrientationJ in ImageJ <http://bigwww.epfl.ch/demo/orientation/>

Response: Thank you for your suggestion, we have applied the color-coding analysis to our multilayer SEM image. It works well with our samples.

- Figure 4: there seems to be variations of thickness in the final samples. Any idea of their origin? Is the shrinkage different depending on the platelet orientation ? Is it due to a wetting of the additional droplet that depend on the platelet orientation in the layer below? What is the maximum number of layers the authors printed? Comments (if any) on the shape control of the final pieces would be welcome.

Response: From our studies, we found that the densification/shrinkage does not depend on the platelet orientations. This is because the packing of the platelets is not dependent on their orientation.

There are two possible sources of the observed variations in thickness. Variations in thickness between different droplets could be caused by the printing equipment, which makes use of pneumatic pressure from a compressed air source to dispense droplets. The pressure of the compressed air source fluctuates during operation and results in slightly different volumes of droplets being dispensed. Another possible source of the observed variation in thickness is from the sample preparation for SEM characterization. For example, the sample in Figure 4c is cut apart with scissors, which causes shearing at the cut location. This may distort the actual structure of the as-printed structures.

In our work, most of the prints have 10 – 20 layers. The largest structure which we printed had up to 50 layers.

- Figure 4: the blue dotted lines are not very visible, maybe try yellow?

Response: We have changed the dotted lines to yellow.

- Figure 5d: orientation based color-coding would be helpful here as well to identify the approximate location of the boundary between successive layers.

Response: We have applied the color-coding analysis here as well. The image has been moved to figure S6

Updated Figure 5:

Figure S6: Sample fabricated for mechanical testing. (a) Optical image of MDOD printed sintered xirallic sample used for mechanical testing. The different shades correspond to different microplatelet orientation. The lighter portion corresponds to $\theta = 0^\circ$ (horizontally aligned) and the darker portion corresponds to $\theta = 90^\circ$ (vertically aligned). (b) Microstructure of sample at the boundary to show the different microplatelet orientations. (c) and (d) High magnification SEM images of the microstructure in each region, showing that the microstructure is as expected. (e) Optical image of xirallic-PDMS composite for compression test. (f) Electron micrograph showing a crack deflection event in a multilayer xirallic-epoxy composite with varying alignments. The colour code corresponds to microplatelets with varying alignment angles.

- methods: is the cleaning of the glass substrates critical for the process? sometimes it can impact the wetting properties, which seems to be critical in this case (Fig 2b)

Response: Yes, the cleaning of the substrate is important for the process. Most of the work uses copper foil as the substrate and they were always cleaned with ethanol before use. As you have mentioned, uncleaned substrates tend to have larger variations in the contact angle and sometimes cause defects in the droplets. We have added a short description about this under Materials and Methods.

Addition:

3D Printing

3D printing was performed using an automated fluid dispensing system (Nordson 3-Axis PROPlus). The ink was filled into 5 ml syringes (Nordson) before loading into the printer. A 0.33 mm inner diameter stainless steel flat tip needle (Nordson) was then installed onto the printhead. The input air pressure was set at approximately 0.5 bar. The copper substrates were prepared by attaching copper foil (Sunhayato) onto glass slides (VWR) to keep them flat. Glass cover slips (VWR) were used as glass substrates. The gold-sputtered substrates were prepared by sputtering (Joel JFC-1600) the glass cover slips with ~10 nm of gold. The substrates were cleaned with ethanol prior to printing to ensure consistent printing as impurities on the substrates may lead to a variation in ink contact angle and could cause defects in the print.

- indications about the typical processing times and dimensions of the pieces would be nice to add to the paper

Response: We have compiled a table of samples' dimensions and processing times in the supplementary information.

Addition to Supplementary Table:

Table S2: Typical processing times and dimensions for various samples

Sample	Shape	Dimensions	Optimised Processing Time
Checkered Sample (Fig. 1)	Cuboid	Length: 1.2 cm Breadth: 1.2 cm Height: 5.5 mm	~6.67 hours
Multilayer Sample (Fig. 4b)	Cylindrical	Diameter: 4 mm Height: 1.5 mm	~30 min
Voxelated Sample (Fig. 4c)	Cuboid	Length: 6 mm Breadth: 4 mm Height: 0.5 mm	~10 min
Nanoindentation test sample (Fig. S6a)	Cuboid	Length: 8 mm Breadth: 8 mm Height: 1.5 mm	~1.5 hours
Compression test sample (Fig. S6e)	Cylindrical	Diameter: 4 mm Height: 3 mm	~1 hour

Pressure Sensor	3 x 3 Array of cylindrical pillars	Length: 1.0 cm Breadth: 1.0 cm Height: 1.5 mm	~45 min
-----------------	------------------------------------	---	---------

- sensors (last part of the paper): although the results seem convincing, I have no particular expertise on sensors so I cannot really comment on this. I nevertheless appreciate the authors efforts to go all the way to demonstration devices.

Response: Thank you for your kind comment.

REVIEWERS' COMMENTS

Reviewer #1 (Remarks to the Author):

In my first review, I submitted an extensive and demanding list of required studies, explanations, and clarifications. The authors have addressed each comment carefully and satisfactorily. I support publication.

Reviewer #2 (Remarks to the Author):

There is not much to say at this point. The authors put considerable efforts and performed additional analysis/experiments to address all the points raised during the review. The figures look great, the arguments are convincing. Not all problems are solved but this is a nice step forward.

I wonder if table S2 should be placed in the paper rather than in the SI, the informations provided in the table are useful and not often reported. I have no other comments at this point.